# *Legionella* effector MavC targets the Ube2N~Ub conjugate for noncanonical ubiquitination

Kedar Puvar [1,4], Shalini Iyer[1,4], Jiaqi Fu[2], Sebastian Kenny[1], Kristos I. Negrón Terón [1], Zhao-Qing Luo [2], Peter S. Brzovic[3,4], Rachel E. Klevit[3✉] & Chittaranjan Das [1✉]

The bacterial effector MavC modulates the host immune response by blocking Ube2N activity employing an E1-independent ubiquitin ligation, catalyzing formation of a γ-glutamyl-ε-Lys (Gln40[Ub]-Lys92[Ube2N]) isopeptide crosslink using a transglutaminase mechanism. Here we provide biochemical evidence in support of MavC targeting the activated, thioester-linked Ube2N~ubiquitin conjugate, catalyzing an intramolecular transglutamination reaction, covalently crosslinking the Ube2N and Ub subunits effectively inactivating the E2~Ub conjugate. Ubiquitin exhibits weak binding to MavC alone, but shows an increase in affinity when tethered to Ube2N in a disulfide-linked substrate that mimics the charged E2~Ub conjugate. Crystal structures of MavC in complex with the substrate mimic and crosslinked product provide insights into the reaction mechanism and underlying protein dynamics that favor transamidation over deamidation, while revealing a crucial role for the structurally unique insertion domain in substrate recognition. This work provides a structural basis of ubiquitination by transglutamination and identifies this enzyme's true physiological substrate.

[1] Department of Chemistry, Purdue University, West Lafayette, IN 47907, USA. [2] Purdue Institute for Inflammation, Immunology and Infectious Disease and Department of Biological Sciences, Purdue University, West Lafayette, IN 47907, USA. [3] Department of Biochemistry, University of Washington, Seattle, WA 98195, USA. [4]These authors contributed equally: Kedar Puvar, Shalini Iyer, Peter S. Brzovic. ✉email: klevit@uw.edu; cdas@purdue.edu

Protein ubiquitination is a post-translational modification used by eukaryotic organisms to regulate critical cellular processes such as protein quality control, cell cycle progression, DNA repair, autophagy, and immunity[1–4]. The sequential action of three enzymes, an ATP-dependent ubiquitin activating enzyme (E1), a ubiquitin-conjugating enzyme (E2), and a ubiquitin-ligase (E3), work to covalently attach the C-terminal glycine (G76) of ubiquitin (Ub) to target proteins, usually through formation of an isopeptide bond to a lysine side chain. Despite lacking a Ub system of their own, many pathogenic bacteria have evolved enzymes or substrate adaptors with the ability to interact with the Ub-signaling system of their eukaryotic hosts, allowing them to take control of host processes and modulate them for their benefit[5,6]. Usually injected into their host cytoplasm through specialized secretion systems, various bacterial effectors have been found to use an array of strategies to hijack or exploit Ub-signaling pathways. Numerous effectors have been found to function as E3 ligases that utilize the host ubiquitination machinery to target host proteins for Ub modification[7]. Other effectors work as deubiquitinases, proteases that cleave the isopeptide bonds that link Ub to target proteins and reverse Ub signals[8,9]. Some effectors even chemically attack and disable specific components of the eukaryotic ubiquitinating machinery directly[10–12], including covalent alteration of Ub itself[13]. However, in recent years, our understanding of this post-translational modification has been redefined by the discovery of bacterial enzymes that catalyze Ub transfer using strategies that bypass the canonical E1–E2–E3 pathway. This was first demonstrated for the SidE family of *Legionella* effectors that catalyze NAD$^+$-assisted phospho-ribosyl linked ubiquitination of certain host targets[14–17].

*Legionella pneumophila* possesses a large arsenal of effectors with over 300 examples of proteins injected into the host via its Dot/Icm Type IV secretion system. These effectors are critical in allowing *L. pneumophila* to form a replicative niche within the host cell where it can survive and avoid host defense mechanisms[18–20]. The newly discovered *L. pneumophila* effector MavC serves as another fascinating example of an enzyme that targets host Ub-signaling pathways, but works in a manner that is distinctly different from the eukaryotic Ub-transfer machinery[21,22] (Supplementary Fig. 1a–c). Valleau et al.[21] first reported the structure of apo-MavC and described its function as a Ub-specific deamidase that catalyzes the conversion of Ub to its Glu40 variant. Together with structural analysis of the effector, the authors concluded that MavC is a structural and functional homolog of known bacterial deamidases such as Cif and CHBP[13,23,24] that target the conserved Gln40 of NEDD8 and Ub. The deamidase structural core in these effectors is conserved in MavC, but MavC also features a unique "insertion" domain that is required for interaction with the E2 Ube2N (also known as Ubc13). These observations led the authors to propose the thioester-linked Ube2N–Ub conjugate as the deamidation substrate to disrupt Ube2N-dependent synthesis of K63-linked Ub chains critical for the innate immune response. However, no clear demonstration of activity on this specific substrate was provided.

Subsequently, Gan et al.[22] co-transfected mammalian HEK293 cells with MavC and a Ub variant that lacks its C-terminal glycine residues and hence cannot be activated or transferred via the canonical eukaryotic E1–E2–E3 pathway. In this context, MavC was observed to modify Ube2N with the Ub variant via a transglutamination reaction that proceeds via an obligate thioester enzyme intermediate between Q40$^{Ub}$ and C74$^{MavC}$. The result is the creation of an isopeptide linkage between the γ-carbonyl group of Gln40$^{Ub}$ and ε-amino group of Lys92$^{Ube2N}$ (Fig. 1a). Ubiquitination of Lys92$^{Ube2N}$, located adjacent to the E2 active site, effectively inhibits Ube2N activity and attenuates downstream host NF-κB activation[25]. Furthermore, the activity

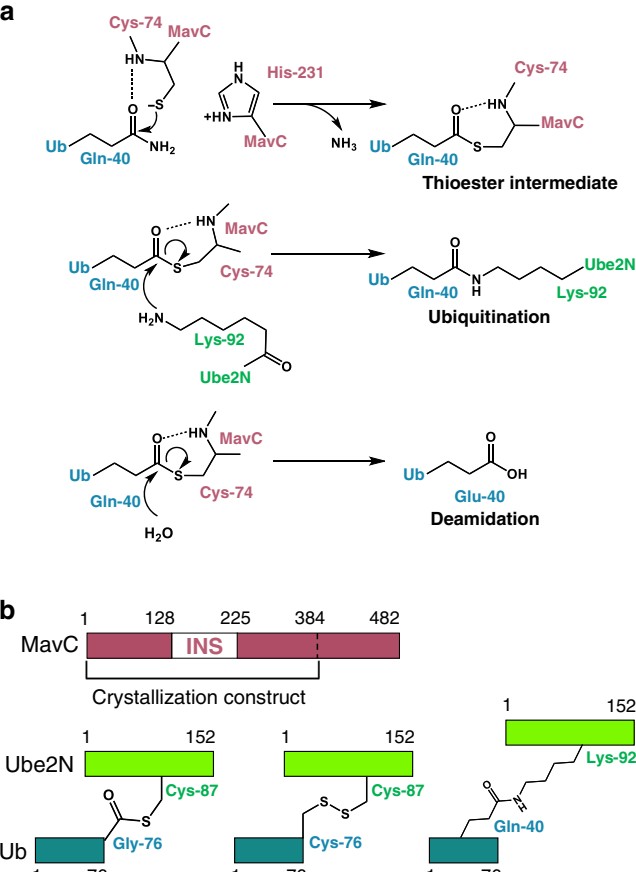

**Fig. 1 Proposed mechanisms of MavC-catalyzed reactions and constructs used for structural studies. a** Proposed mechanism of ubiquitination resulting from transglutamination reaction catalyzed by MavC. Thioester formed in Step 1 may undergo either attack by the amino group of Ube2N Lys92 (leading to formation of Ub–Ube2N) or hydrolysis (formation of deamidated Ub). Key residues from MavC in burgundy, Ube2N in light green, and Ub in teal are represented. **b** Diagram depicting protein constructs used for crystallization studies and NMR experiments and location of the MavC insertion domain (residues 128–225), with diagrams of Ube2N-Ub, Ube2N-SS-Ub (MavC substrates), and Ub–Ube2N (MavC product) provided for comparison.

of MavC is antagonized by MvcA, a protein of 50% identity with MavC that functions to remove Ub from Ub–Ube2N in later phases of infection[26], which points to the importance of temporal regulation of the activity of this E2 enzyme during *L. pneumophila* infection. Thus, MavC catalyzes what seems to be the only known example of a Ub transfer reaction that does not require a nucleotide cofactor to activate Ub prior to substrate modification[22].

A key assumption of the previous work was that MavC recognizes and joins free Ub and Ube2N together[22]. However, in cells it has been predicted that E2s exist predominantly as the activated E2–Ub conjugate poised to transfer Ub to substrates[27]. Moreover, while MavC-catalyzed Ub deamidation occurs fairly slowly compared to the transglutaminase-mediated E2 ubiquitination activity[22], it remains unclear how the enzyme prioritizes one activity over the other given their mutual exclusivity. Here, we demonstrate that MavC actually targets the Ube2N–Ub conjugate to catalyze an intramolecular transglutamination reaction. We present a series of crystal structures of MavC in complex with a disulfide-linked substrate that mimics the charged Ube2N–Ub conjugate and with the transglutaminase crosslinked product

(Fig. 1b). These structures reveal how interaction with MavC leads to remodeling of regions surrounding the Ube2N active site to promote the intramolecular transglutaminase reaction and inhibit Ube2N–Ub function. The progression of structures reveals key features in MavC that provide a basis for understanding its transglutaminase mechanism, substrate specificity, and how conformational dynamics favor a specific reaction outcome.

## Results

**MavC efficiently transglutaminates a Ube2N–Ub mimic**. Previous work on MavC implicated free Ube2N and/or Ub as substrates[21,22]. To determine how MavC engages substrates we used biolayer interferometry (BLI) to measure binding affinities and NMR to map MavC/substrate interactions. The combined results show that free Ub binds only weakly to MavC and binding constants could not be determined with high confidence (Fig. 2a). Though an approximate $K_d$ of ~170 μM for binding of free Ub to MavC was deduced from BLI measurements, reaction assays and NMR experiments suggest the affinity is much weaker: reaction assays monitoring Ub deamidation as a function of Ub concentration failed to saturate activity even at 375 μM Ub, suggesting that the $K_M$ for free Ub in this reaction is in excess of 200 μM (Fig. 2b, Supplementary Fig. 2a, b). NMR titration of $^{15}$N-Ub using a catalytic Cys-to-Ala mutant of MavC (C74A-MavC$_{1–384}$) revealed almost no chemical shift perturbations (CSPs) in the Ub spectrum at equimolar (150 μM) concentrations. Further addition of 150 μM Ube2N to the NMR sample did not appreciably enhance the weak interaction between MavC and Ub (Fig. 2c). We note that a previous study conducting similar NMR titration experiments, but using catalytically active MavC, reported significant CSPs in Ub upon addition of MavC[21]. However, we found those results consistent with the MavC-catalyzed conversion of Ub$_{WT}$ to the deamidated product, Ub$_{Q40E}$, and not with direct protein–protein interactions (Supplementary Fig. 2c, d).

In marked contrast to Ub, Ube2N binds MavC with much higher affinity. Indeed, previous work identified a MavC/Ube2N complex in cell extracts[21]. Our BLI measurements yielded a $K_d$ of ~2.5 μM for Ube2N (Fig. 2a). In NMR titration experiments, numerous CSPs are observed upon addition of C74A-MavC$_{1–384}$ to $^2$H,$^{15}$N-labeled Ube2N (Supplementary Fig. 2e). The observed CSPs define a MavC recognition surface on Ube2N formed by residues in Helix1, Loop4, and Loop7 (Supplementary Fig. 2f). This is the same Ube2N surface shown to interact with numerous eukaryotic E3 ligases[28]. An analogous NMR titration using purified MavC insertion domain (MavC$_{128–225}$) yielded Ube2N spectral changes that are remarkably similar to those observed with C74A-MavC$_{1–384}$ despite the large molecular weight differences (Supplementary Fig. 2g). These observations suggest that the MavC insertion domain is primarily responsible for binding Ube2N, a conclusion supported by similar $K_d$ values obtained from BLI measurements of Ube2N binding to MavC or the MavC insertion domain (Fig. 2a, Supplementary Fig. 1e).

Despite the relatively high affinity for Ube2N, the weak interaction between MavC and Ub ($K_M > 200$ μM) indicates an intermolecular transglutamination reaction between free Ub and Ube2N would be unlikely to occur under cellular conditions. As most cellular E2s are predicted to have Ub tethered to its active-site Cys via a thioester linkage[27,29], we set out to assess whether Ube2N–Ub is the relevant substrate for MavC. Our approach utilized a stable Ube2N–Ub mimic[30] in which the G76C$^{Ub}$ mutant is disulfide linked to the active site Cys87$^{Ube2N}$ (Fig. 1b). This mimic provided greater control of reaction components and minimized experimental complications that would arise from hydrolysis of purified wild-type Ube2N–Ub during MavC reaction assays and during NMR experiments. (Hereafter, this substrate surrogate is referred to as Ube2N-SS-Ub or the disulfide conjugate.)

BLI and NMR experiments show MavC readily binds Ube2N-SS-Ub. BLI-binding titrations yield a $K_d$ of ~2.4 μM for Ube2N-SS-Ub, nearly the same affinity as observed for free Ube2N (Fig. 2a), consistent with binding dominated by interactions with Ube2N. NMR experiments were performed using a disulfide conjugate in which only the Ub subunit was isotopically labeled (Ube2N-SS-$^2$H,$^{15}$N-Ub). In the conjugate, not all Ub resonances are observed or of equal intensity. This is due to the Ub subunit alternating between open states, where Ub makes few contacts with the E2, and closed states, where the Ub subunit is in close contact with the E2 (ref. [31]). This equilibrium results in exchange broadening of resonances whose environments differ in the open and closed states. Ub resonances that remain and largely overlap with those of free Ub can be assigned by inspection (Supplementary Fig. 2h, i).

In marked contrast to the addition of MavC to free Ub (Fig. 2c), large perturbations in the spectrum of the Ub subunit of $^2$H,$^{15}$N-Ub-SS-Ube2N are now observed upon formation of an $^2$H,$^{15}$N-Ub-SS-Ube2N/MavC complex (Fig. 2d). An overall loss in peak intensity is observed consistent with the large increase in molecular weight (~65 kDa) upon complex formation. In addition, a number of Ub resonances disappear. Again, this behavior can be attributed to resonance exchange broadening where a subset of Ub residues exchange between contacts with MavC, Ube2N, and/or solvent. Thus, in solution, the Ub subunit is not rigidly fixed to the enzyme active. However, the high local concentration of Ub provided by MavC binding of the Ube2N–Ub conjugate significantly increases observed contacts.

In targeting Ube2N–Ub for modification, MavC could catalyze deamidation of the Ub subunit, an intramolecular transglutaminase reaction between the E2 and Ub subunits, or some combination of the two. In all scenarios, tethering the C-terminus of Ub to the E2 active site must not hinder the ability of MavC to form an obligate thioester intermediate with the Gln40 side chain of the Ub subunit. Furthermore, to catalyze transglutamination, MavC must be able to orient Lys92$^{Ube2N}$ and Gln40$^{Ub}$ of the E2–Ub conjugate in proximity to form an isopeptide bond. To investigate these possibilities, we conducted assays using 25 μM Ube2N-SS-Ub conjugate as substrate and enzymatic amounts of MavC (5 nM). Under these conditions, MavC exhibits robust transglutaminase cross-linking activity, while deamidation of Ub of the disulfide substrate was not detected (Fig. 2e, Supplementary Fig. 2k). In sharp contrast, reactions using free Ub and Ube2N at the same subunit concentrations as the disulfide conjugate (25 μM) produced no detectable transglutaminase product (Fig. 2e). These results strongly argue that MavC targets the Ube2N–Ub conjugate to catalyze an intramolecular transglutaminase reaction resulting in the formation of an isopeptide bond between Ube2N and Ub. These results are also in line with observations of Gan et al.[22] which show that a mutant of Ub lacking the last two glycines is modified to a significantly lower extent than wild-type Ub.

**Structural basis of transglutaminase-mediated ubiquitination**. MavC-catalyzed transglutamination proceeds via an obligate thioester-linked intermediate to form an isopeptide bond between Gln40$^{Ub}$ and Lys92$^{Ube2N}$ (Fig. 1a). To gain structural insights into the mechanism underlying this noncanonical ubiquitination, we sought to crystallize MavC with both substrate and product. For crystallization trials, we used a truncated MavC construct, MavC$_{1–384}$ (Fig. 1b), that otherwise retains full enzymatic activity and Ube2N binding (Supplementary Fig. 1d, e). In addition, the MavC active site residue, Cys74, was mutated to Ala (C74A) to

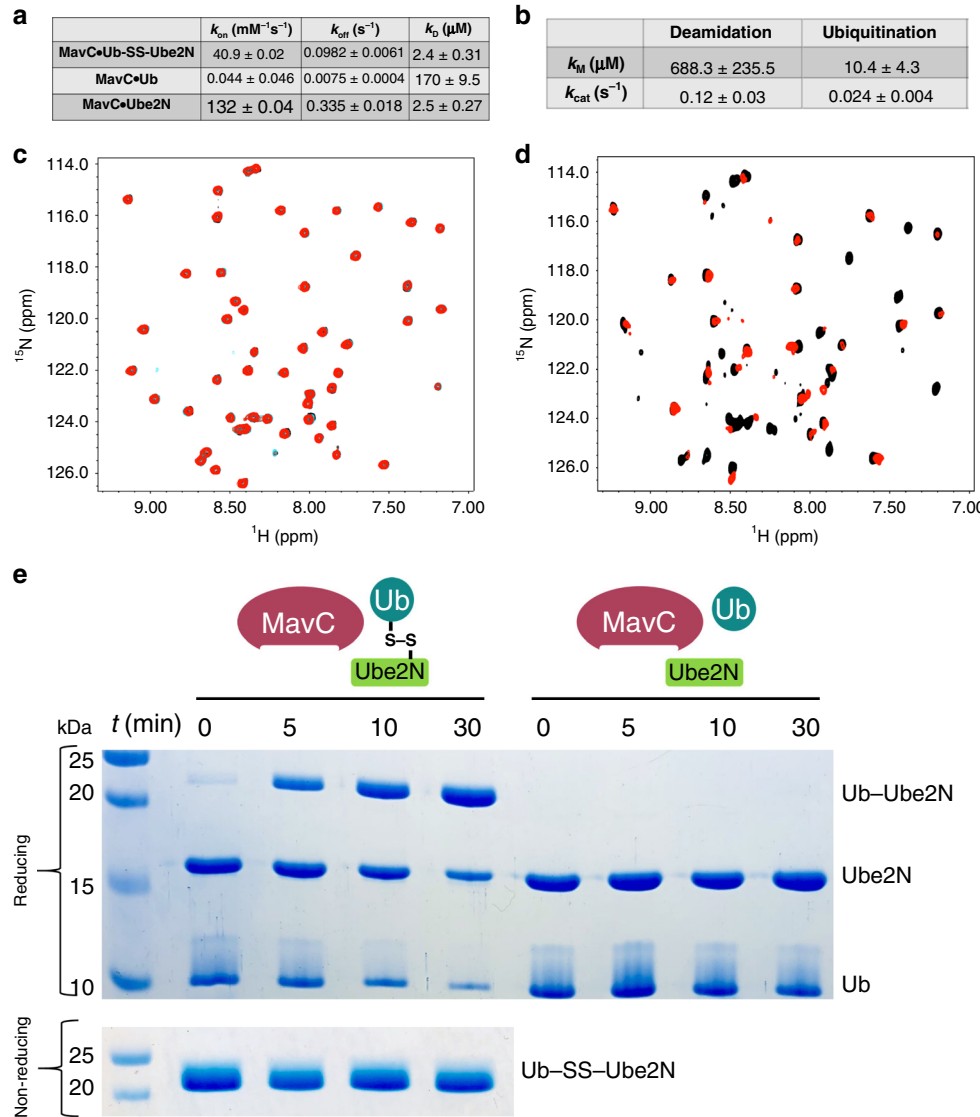

**Fig. 2 Ube2N-SS-Ub is a significantly better substrate for MavC than free Ub/Ube2N. a** Biolayer interferometry table showing the measured on and off rates and the dissociation constants of MavC (C74A) with Ube2N-SS-Ub, Ub, and Ube2N. **b** Michaelis–Menten parameters of ubiquitination (with respect to Ube2N) and deamidation (with respect to Ub) reactions. **c** Free Ub binds weakly to MavC. $^1$H,$^{15}$N-HSQC NMR spectral overlays of 150 μM $^{15}$N-Ub alone (black spectrum) or in the presence (cyan spectrum) of 150 μM C74A-MavC$_{1-384}$. CSPs are weak or negligible indicating that any interaction between MavC and free Ub is very weak. Addition of 150 μM Ube2N to the sample (red spectrum) has marginal effects on the Ub spectrum, and therefore, does not significantly change Ub binding to MavC at NMR concentrations. **d** Ube2N-SS-Ub binds to MavC. Spectral overlay of 150 mM Ube2N-SS-$^2$H,$^{15}$N-Ub alone (black spectrum), or in the presence of 150 μM C74A-MavC$_{1-384}$ (red spectrum). Only the Ub subunit is isotopically labeled. The Ub subunit was $^2$H-labeled to allow for NMR observation in high molecular weight complexes. Select Ub subunit resonances are observed to shift or disappear indicating that the Ub subunit interacts with MavC. **e** A time course comparison of the ubiquitinating activity of MavC on Ube2N-SS-Ub versus free Ube2N and Ub as the substrates. Loading controls for Ube2N-SS-Ub for each time point under non-reducing conditions are also shown. Reactions were subjected to SDS-PAGE and visualized with Coomassie Blue. A control reaction at time 0 is included. Accompanying each time course is a cartoon representation of the reaction components.

prevent any modification of Ub or Ube2N during crystallization. The MavC/substrate complex was generated using the substrate mimic Ube2N-SS-Ub used in NMR studies and in biochemical assays (see Fig. 2). The substrate complex crystallized in three different space groups (Table S1), C222, R3, and P6$_5$. The product complex was generated using Ub–Ube2N product isolated from a MavC-catalyzed reaction mixture where free Ube2N and Ub were linked via a γ-glutamyl-ε-Lys isopeptide link.

The structure of MavC$_{1-384}$ in all complexes (Fig. 3) is topologically identical to that in the previously determined structure of apo-MavC$_{1-384}$ (PDB id 5TSC[21]). Architecturally, MavC$_{1-384}$ is composed of three distinct lobes: a core globular

domain (CG) flanked by an insertion domain (residues 128–225) on one end and an α-helical extension (HE: residues 33–66, 356–384) on its opposite end. The overall structure of MavC$_{1-384}$ can be described as a C-shaped crescent with a crevice where the catalytic center is located (Fig. 3). Looking down into the concave face of the crescent, the MavC catalytic triad (residues Cys74 (C74A), His231, and Gln252) is located near the center of the C-shaped structure (Fig. 3). In all structures, the Ube2N and Ub subunit bind in an extended conformation, with Ube2N on one side of the active site and Ub on the other. The Ub subunit binds between the insertion domain and the HE, positioning the Gln40$^{Ub}$ side chain within the MavC catalytic cleft. The structures

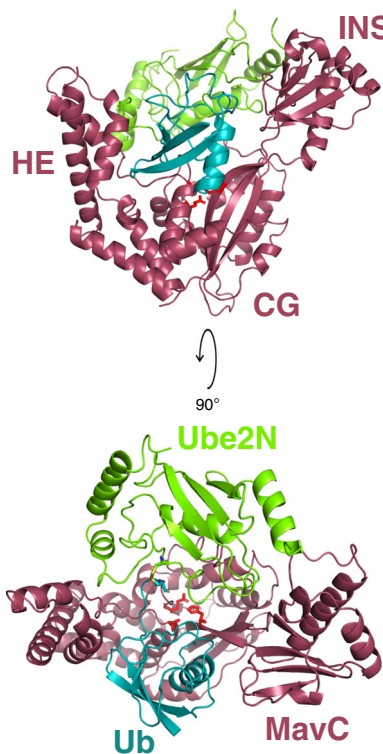

**Fig. 3 Overall structure of Ube2N-SS-Ub bound to MavC.** Cartoon representation of the crystal structure of MavC•Ube2N-SS-Ub complex (P6₅), with MavC depicted in burgundy, Ube2N in green, and Ub in teal. Key domains (CG, core globular domain; HE, helical extension; INS, insertion domain) are labeled. Rotated view depicting a top-down view of the active site is also shown. MavC active site residues C74, H231, and Q252 are depicted as red sticks.

are consistent with our solution NMR binding experiments. The surface of the Ube2N subunit formed by Helix1, Loop4, and Loop7 makes extensive contacts with the MavC insertion domain (Fig. 3, Supplementary Fig. 2f). Observable Ub resonances not significantly affected by complex formation are very similar to those of free Ub (Supplementary Fig. 2h, i). These resonances can be assigned by inspection and are primarily in located in regions of the Ub subunit that do not make contact with MavC (Supplementary Fig. 2j).

Though the Ub subunit does not bind tightly to MavC in solution, the structures reveal a number of potential contacts that we found important for activity. Residues in the MavC HE region (MavC Leu36, Asn39, Glu40, Ile43, and Glu66) interact with the N-terminal β-hairpin turn of Ub as well as the nearby C-terminal Ub tail (Ub Leu8, Thr9, Arg72, and Arg74) (Fig. 4a, b). Around the MavC catalytic triad, numerous contacts are observed, including a rare contact with the di-Pro motif of Ub (Pro37–Pro38) using a CH–O hydrogen bond between the carbonyl group of Ala229$^{MavC}$ and Hδ atoms of the Pro38$^{Ub}$ (ref. [32]) (Fig. 4b). Numerous hydrophilic contacts include Asn72$^{MavC}$ positioned to H-bond with the backbone carbonyl of Leu73$^{Ub}$, Arg121$^{MavC}$ in H-bonding distance to the carbonyl of G35$^{Ub}$, and the side chains of R126$^{MavC}$ and Thr230$^{MavC}$ poised to interact with the Asp39$^{Ub}$ carboxylate. Mutation of these MavC residues leads to a reduction or complete loss of the ability of MavC to catalyze either transglutamination (using the disulfide conjugate as a substrate) or Ub deamidation (Fig. 4c, d). Mutation of Arg72$^{Ub}$, one residue that distinguishes Ub from the structurally related NEDD8, to alanine severely impairs the ability

of MavC to recognize this mutant substrate (Supplementary Fig. 3a).

Modeling a Cys side chain in a preferred rotamer orientation in place of MavC Ala74 reveals the γ-S atom poised for nucleophilic attack at Gln40$^{Ub}$, approaching within 3.0 Å of the carboxamide group (Fig. 4e). Gln40$^{Ub}$ is held in the active site with its side chain C=O group pointing toward the backbone amide groups of the catalytic Cys74$^{MavC}$ and Trp255$^{MavC}$ (H-bonding distance of 2.9 and 3.0 Å, respectively) whereas the NH₂ group is in H-bonding contact with the backbone carbonyl group of Thr230$^{MavC}$ (2.8 Å) and the imidazole side chain of His231 (Fig. 4e). The backbone amides would be important for stabilization of the oxyanion transition state and the His231 interaction is likely for proton donation to the leaving ammonia during the thioester step (Fig. 1a). The indole side chain of Trp255$^{MavC}$ is stacked against the Gln40-Gln41 peptide unit of Ub (Fig. 4e), an arrangement that permits the backbone carbonyl of Gln40$^{Ub}$ to come within H-bonding distance from the hydroxyl group of Ser73$^{MavC}$. Mutation of Trp255 and Ser73 of MavC to alanine results in a significant loss of Ub deamidation and transglutamination activity of the enzyme (Fig. 4c, d). Combined, these interactions appear to fix the Gln40$^{Ub}$ side chain in a reactive arrangement for attack by the nucleophilic Cys to facilitate formation of the thioester intermediate. In this arrangement the NH₂ group points toward a solvent-filled area, which would allow the ammonia produced during catalysis to diffuse away from the active site.

In both substrate and product complexes, there are three regions of MavC that interact with Ube2N. Region 1, the MavC insertion domain, makes an overwhelming contribution to Ube2N binding compared to other parts of MavC. Around 500 Å (ref. [2]) of the surface area is buried at the interface between MavC$_{INS}$ and Ube2N alone. Accordingly, its deletion results in a dramatic loss of Ube2N binding and no detectable ubiquitination activity (Fig. 5d)[21]. The MavC insertion domain can be expressed and purified on its own, and a crystal structure of the insertion domain shows that it preserves an identical fold to that in the full MavC protein (Supplementary Fig. 4a). BLI and glutathione-s-transferase (GST) pulldown experiments show that the isolated insertion domain is able to bind to Ube2N independently and with an affinity comparable to full-length MavC (Supplementary Fig. 4b, c). The second Ube2N-interacting region of MavC corresponds to residues in the CG domain, particularly Met317$^{MavC}$, that interacts with Loop4, the 3₁₀-helix containing Lys92$^{Ube2N}$, and αHelix2 of Ube2N (Fig. 5b). The third region involves contacts between the MavC HE domain and Ube2N αHelix3 (Fig. 5c), an interface largely supported by a network of polar contacts between acidic and basic residues from both MavC and Ube2N. The contacts in this region appear to play a key role as the substrate transitions through the catalytic process. For example, interactions involving Arg63$^{MavC}$ and Lys64$^{MavC}$ with the αHelix3 of Ube2N are observed only in the P6₅ substrate complex structure, which is closer to the product complex (also in P6₅) than the other substrate complexes. Consistent with these observations, mutation of Arg63$^{MavC}$, Lys64$^{MavC}$, and Glu66$^{MavC}$ in the HE, Phe188$^{MavC}$, Tyr189$^{MavC}$, Tyr198$^{MavC}$, and Glu207-$^{MavC}$ in the insertion domain, and Met317$^{MavC}$ in the CG severely impair MavC catalytic activity (Fig. 5d). Tyr47$^{MavC}$ was also chosen as a site for mutation in these mutants because it contributes to hydrophobic interactions at the HE interface. Based on results from the biochemical experiments, we examined the effects of a Tyr47/Tyr198/Glu207$^{MavC}$ to alanine triple mutant (the YYE mutant) on MavC-induced Ube2N ubiquitination in cells infected by *L. pneumophila*. We also created mutants by additionally changing either Met317 or Trp255 to alanine in

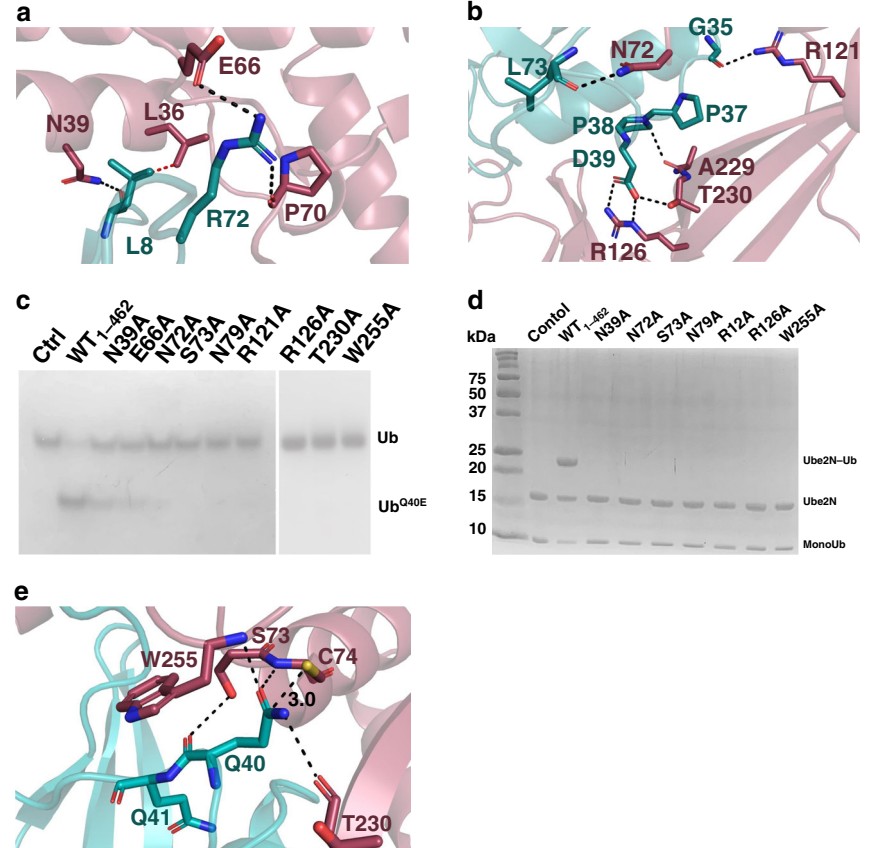

**Fig. 4 Binding interface of MavC with Ub. a, b** Views of the MavC•Ub-binding interface showing interactions between residues of MavC (burgundy) and Ub (teal). Hydrogen-bonding interactions are shown in black dashed line and hydrophobic contacts in red dashed line. **c** Native-PAGE gel comparing Ub deamidation activities of Ub-binding mutants of MavC. Reactions were visualized with Coomassie Blue. A control reaction without MavC is included. **d** Comparison of the ubiquitinating activity of the Ub-binding mutants of MavC using Ube2N-SS-Ub as the substrate. Reactions were subjected to SDS-PAGE and visualized with Coomassie Blue. A control reaction without MavC is included. **e** Architecture of the active site in the structure of MavC substrate complex (P6$_5$), with Ala74 of MavC modeled as a Cys. Interactions holding Gln40$^{Ub}$ in position are described.

the YYE mutant (YYE/M317A, YYE/W255A). Unlike the single point mutants or YYE, which show appreciable activity under longer reaction times in the transglutaminase assay, these quadruple mutants were much more defective (Supplementary Fig. 3b, c). All of these mutants were translocated into infected cells at levels comparable to that of the wild-type complementation. We then examined the levels of Ube2N ubiquitination in infected cells. We also examined the ability of MavC and the mutants to attenuate NF-κB activation under conditions of TRAF6 overexpression[22]. Whereas the activity of the YYE mutant has severely impaired the ability to modify Ube2N and displays defects in attenuating NF-κB activation, the quadruple mutants were further impaired to levels comparable to the catalytically inactive C74A mutant, in line with the biochemical activity assay results (Fig. 5e, f, Supplementary Fig. 3c). Collectively these results indicate that the transglutaminase activity of MavC targeting the Ube2N–Ub conjugate is essential for its biological role in attenuating NF-κB response in host cells.

**MavC remodels the Ube2N active site to promote crosslinking**. Remarkably, comparison of substrate and product MavC complexes reveal a progression in the conformation of loops surrounding the Ube2N active site. In one structure containing Ube2N-SS-Ub (Fig. 6a), the E2 conformation is very similar to 49 other structures of Ube2N available in the Protein Data Bank

(Supplementary Fig. 4d). The Ube2N active site loop formed by residues 116–123 is the most variable region among Ube2N structures, and in the MavC complex it adopts an altered conformation relative to an average Ube2N structure (Supplementary Fig. 4d). Lys92$^{Ube2N}$, the residue that will form an isopeptide bond with Gln40$^{Ub}$, is located in a 3$_{10}$-helix that is common to all other Ube2N structures. Though the complete Lys92$^{Ube2N}$ side chain is not resolved in this structure, its β-carbon is positioned over 16 Å away from the γ-carbon of Gln40$^{Ub}$. This placement is too far for reaction with a thioester intermediate and isopeptide bond formation, suggesting that a conformational change is required to bring Lys92$^{Ube2N}$ into the MavC active site. The other substrate and product MavC complexes reveal a dramatic change in the conformation of the 3$_{10}$-helix that brings the Ube2N target lysine into position to attack the γ-carbon of Gln40$^{Ub}$ (Fig. 6a–c show the three states we call "early substrate", "attacking substrate," and "product", respectively, with respect to the transition of the 3$_{10}$-helix). A methionine in MavC, Met317$^{MavC}$, appears to play a critical role in promoting this conformational change. In the early substrate/MavC structure (Fig. 6a), Met317$^{MavC}$ is positioned just below Ube2N Helix-2 and adjacent to the 3$_{10}$-helix. In the attacking substrate/MavC complex, the electron density for the 3$_{10}$-helix is lost suggesting this region adopts multiple conformations in the crystal. Here, Met317$^{MavC}$ has shifted ~4.5 Å relative to the E2 into a hydrophobic pocket of the E2 formed by the movement of the 3$_{10}$-helix. In the product

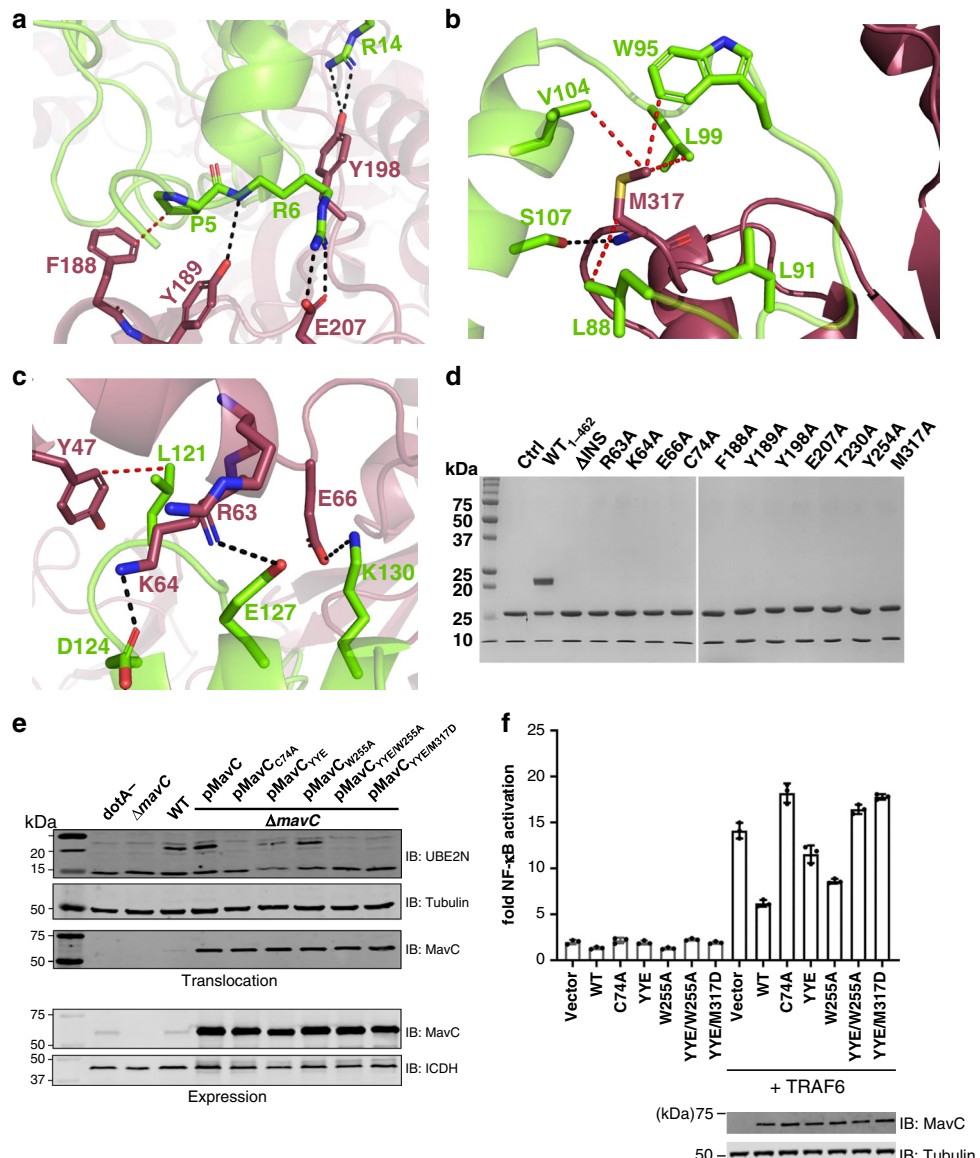

**Fig. 5 Binding interface of MavC with Ube2N. a–c** Detailed view of interactions in MavC regions 1, 2, and 3 with Ube2N. Key residues are labeled and represented as stick models. MavC is depicted in burgundy and Ube2N in green. Hydrogen-bonding interactions are given in black and hydrophobic interactions in red. **d** Comparison of the ubiquitinating activity of wild-type MavC versus mutant proteins using Ube2N-SS-Ub as the substrate. Reactions were subjected to SDS-PAGE and visualized with Coomassie Blue. A control reaction without MavC is included. **e** MavC-mediated ubiquitination of Ube2N during L.p infection. Cells infected with the indicated *L. pneumophila* strains were lysed with 0.2% saponin and lysate separated by SDS-PAGE, probed by immunoblotting with antibodies specific for Ube2N (upper panel) and MavC (lower panel). respectively. **f** The effects of MavC and its mutants on NFκB activation. HEK293T cells were transfected with plasmids expressing a luciferase reporter responsive to NF-κB and Flag-MavC or its mutant. At the same time, a plasmid expressing *Renilla* luciferase used as an internal control and stimulator TRAF6 were co-transfected. NF-κB activity was determined by dual luciferase assay. The expression of MavC and its mutants was probed in lysates of transfected cell while tubulin was detected as a loading control. Three independent experiments were done with similar results. Error bars indicate standard error of the mean (SEM).

complex, the $3_{10}$-helix is highly extended, with Lys92$^{Ube2N}$ positioned in the MavC active site covalently linked to Gln40$^{Ub}$ via an isopeptide bond. Notably, the methyl group of Met317$^{MavC}$ occupies the position vacated by Ile90$^{Ube2N}$ as the $3_{10}$-helix unfolds. Consistent with a role in stabilizing the extended Ube2N conformation, mutation of Met317$^{MavC}$ abrogates MavC trans-glutaminase activity (Fig. 5d).

Additional interactions may be important for stabilizing the extended structure of Ube2N. In the product complex, the aliphatic portion of the Lys92 side chain is held in place by van der Waals interactions with Tyr254$^{MavC}$ (Fig. 6c, inset) positioning the ε-amino group within 4.2 Å from Cβ of Ala74. The proximity

of the backbone carbonyl of Thr230$^{MavC}$ to Lys92 also indicates a potential hydrogen bond that may stabilize the Lys amino group in a productive orientation. Though the side chain of Lys92$^{Ube2N}$ is well-defined in the product complex, the region of Ube2N surrounding this residue still appears to be dynamic. This is inferred from weaker electron density and the relatively high average B-factor of ~70 Å$^2$ for the residues from 86 to 95, compared to ~40 Å$^2$ for the whole complex. Thus, it appears that interactions between Ube2N–Ub and MavC lead to large conformational changes in regions surrounding the Ube2N active site, especially unfolding of the Ube2N $3_{10}$-helix, which is necessary to position Lys92$^{Ube2N}$ into the MavC active site.

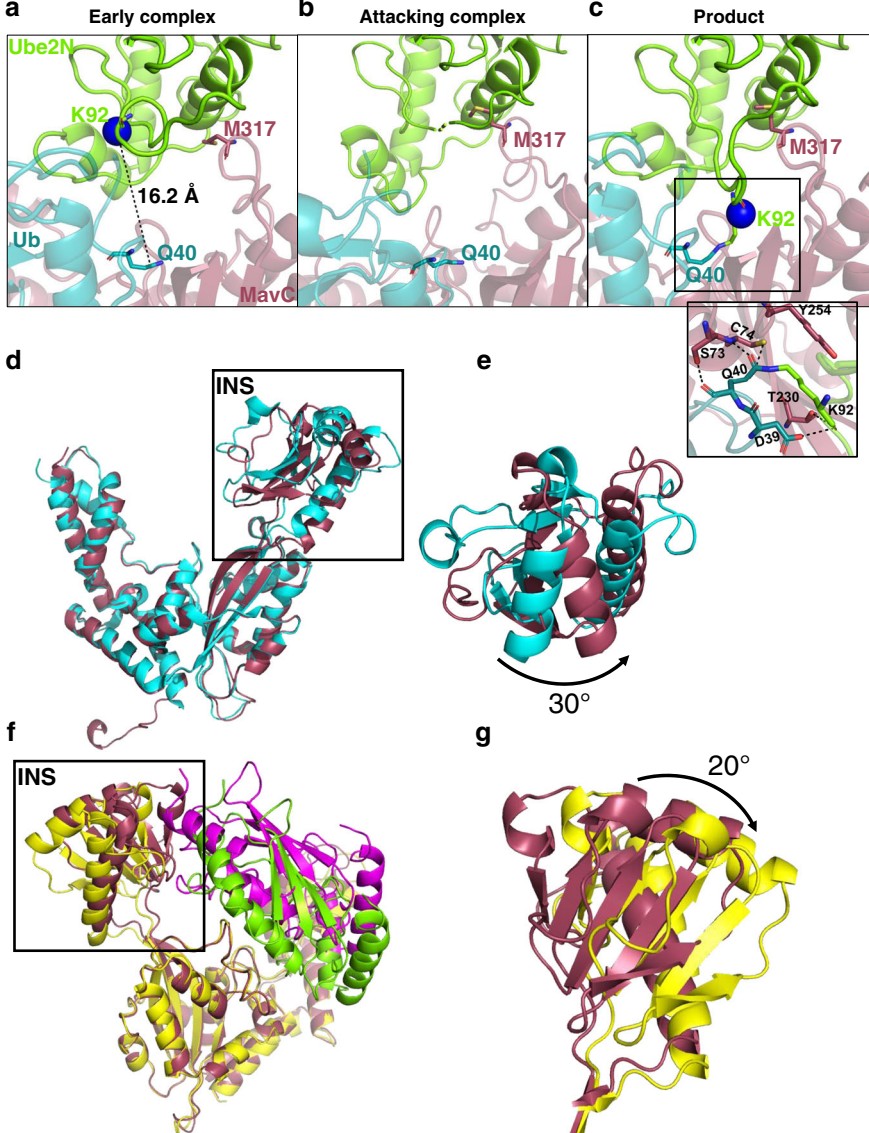

**Fig. 6 Conformational dynamics of MavC and remodeling of the Ube2N active site.** Superposition of Ube2N in three different structures (**a–c**) reveal the remodeling of Ube2N loops that occurs upon interaction with MavC. A comparison of crystal structures of MavC bound to Ube2N-SS-Ub in the P6₅ space group (**a**), in the R3 space group (**b**), and MavC bound to the product Ub–Ube2N (**c**) captures the progression of the 3₁₀-helix of Ube2N from canonically structured to destabilized to unfolded. The β-carbon of the target Lys92$^{Ube2N}$ is highlighted with a blue sphere. Note the changing position of MavC M₃₁₇ that accompanies unfolding of the Ube2N 3₁₀ helix. Inset **c** shows interactions involving Lys92$^{Ube2N}$ in the active site of MavC as observed in the product-bound structure. **d** Structural alignment of apo-MavC (cyan) with substrate-bound MavC (R3, burgundy). Insertion domain is highlighted with a box. **e** Zoomed view of the insertion domain of apo and bound MavC (color coded as previously), depicting the rigid-body rotation upon Ube2N-SS-Ub binding. **f** Structural alignment of the two substrate-bound crystal structures of MavC (R3: MavC in burgundy, Ube2N in green. P6₅: MavC in yellow, Ube2N in magenta). Insertion domain is highlighted with a box. **g** Zoomed view of insertion domain in both substrate-bound crystal forms of MavC depicting the pendulum rotation required to bring Ube2N near active site.

**Conformational dynamics of the MavC insertion domain.** Comparison of the structure of apo-MavC (PDB id 5TSC) to the substrate complexes reveals that Ube2N could not bind to the insertion domain in apo-MavC without steric interference from the MavC HE. However, our NMR titration data show that Ube2N readily binds MavC in solution (Supplementary Fig. 2e, g). This suggests that flexibility of the insertion domain relative to the MavC CG and HE domains are important for function. Indeed, comparison of apo-MavC with substrate-bound complexes reveals that insertion domain undergoes a pronounced 30° rigid-body rotation (Fig. 6d, e) that would enable MavC to accommodate the Ube2N-SS-Ub substrate. As the reaction proceeds further, the insertion domain undergoes a second

rigid-body pendulum movement that helps to bring Ube2N, and the side chain of Lys92$^{Ube2N}$, into the active site (Fig. 6f, g). Thus, MavC catalysis involves binding of Ube2N–Ub to the MavC insertion domain, motion of the insertion domain to allow additional contacts of the E2 and Ub subunits with the MavC HE and CG domains, and remodeling of the Ube2N 3₁₀-helix to position Lys92$^{Ube2N}$ in the MavC active site for isopeptide bond formation.

**MavC can target the Uev1a/Ube2N–Ub conjugate.** Essential for Ube2N-dependent activation of NF-κB is the synthesis of K63-linked poly-Ub chains. This requires complex formation of

Ube2N with an E2 variant protein, such as Uev1a, that binds the acceptor Ub and directs synthesis of K63 linkages. Thus, it is likely that MavC targets Uev1a/Ube2N–Ub conjugates in order to disrupt NF-κB activation. Uev1a binds tightly to Ube2N[33], interacting with residues in β-strands 3 and 4 of Ube2N and some of the interconnecting loops, including the extended loop that leads into the E2 active site. Our structures predict that this interaction site is solvent accessible in the MavC complex, and that MavC should also be able to target the Uev1a/Ube2N–Ub conjugate. Accordingly, we demonstrated that Uev1A can bind to Ube2N-SS-Ub as deduced from co-elution of the two proteins during size-exclusion chromatography (Supplementary Fig. 5a). This prediction is also borne out by in vitro activity data that shows increasing additions of Uev1a to the reaction mixture does not inhibit the ability of MavC to catalyze the intramolecular transglutamination of Ube2N and Ub (Supplementary Fig. 5b). In agreement with these results, our structural analysis reveals that Uev1A binding to Ube2N–Ub is unlikely to interfere with MavC binding (Supplementary Fig. 5c).

## Discussion

MavC catalysis presents a remarkable example of a ubiquitination reaction achieved through transglutaminase chemistry that does not require nucleotide-dependent activation of Ub. The result is isopeptide crosslinking of Ub to a specific target, Ube2N. Though other E2s harbor a structurally equivalent target lysine residue, selectivity for Ube2N is achieved by binding the same interface recognized by cognate eukaryotic E3 ligases (Supplementary Fig. 3b–d). Furthermore, low concentrations of MavC (nM) effectively target and inhibit both activated Ube2N–Ub conjugate and the Uev1a/Ube2N–Ub complex by catalyzing the synthesis of an intramolecular isopeptide bond adjacent to the E2 active site. Catalysis requires extensive remodeling of loops surrounding the E2 active site, structural changes that have not been previously observed in other Ube2N structures. Targeting a specific E2–Ub conjugate allows *L. pneumophila* to modulate specific host cellular processes instead of a systemic effect on the entire host Ub landscape.

With an approximate cellular concentration of 2 μM, Ube2N is among the most abundant E2 enzymes in cells, existing mostly as the thioester-linked Ube2N–Ub conjugate[27,34]. Accordingly, MavC might have evolved to target this specific form of the E2 wherein transglutaminase catalysis would involve intramolecular crosslinking between Ub and E2 subunits of the conjugate. Notably, the binding affinity of MavC for this substrate (Fig. 2a) is within the range of its estimated cellular concentration. As a translocated effector from the phagosome, MavC is likely present at extremely low levels in the host cytosol, where such matching of substrate availability and binding affinity would ensure efficient targeting of the desired substrate. As Ube2N itself is a constitutive part of the enzymatically active heterodimeric complex with Uev1a (or Uev2a), the actual in vivo substrate of MavC is most likely the heterodimeric complex, the Uev1a:Ube2N–Ub complex[35]. In line with this notion we find that Uev1a binding does not interfere in the ubiquitination activity of MavC (Supplementary Fig. 5b). Inactivation of the Uev1a:Ube2N–Ub complex through MavC-mediated Ub crosslinking to the E2 would result in inhibition of synthesis of Lys63-linked poly-Ub chains ultimately affecting NF-kB activation. This inhibition may occur due to blocking of re-charging of Ube2N by the E1 enzyme, and also by competitive displacement of E3 binding (Fig. 7b, Supplementary Fig. 5d, e).

Although first reported to be a Ub-specific deamidase[21], we find that MavC has weak affinity for free Ub, that is far too weak for a meaningful level of Ub deamidation activity under cellular conditions. This may provide an explanation for why deamidated Ub was undetectable in cells infected with wild-type *L. pneumophila*[21,22]. The ubiquitinating machinery of MavC is constructed from a deamidase core that is shared among previously characterized bacterial Ub/NEDD8 deamidases, to which is appended an insertion domain that serves as the E2 recruitment element. The binding affinity for Ube2N, largely contributed by the insertion domain, is relatively high, perhaps enabling MavC to compete with host E3 ligases that recruit the same E2. We propose that by targeting Ube2N–Ub the enzyme leverages high-affinity interactions with the E2 subunit to effectively increase the local Ub concentration, thereby circumventing its low affinity for free Ub. Ub is seen nestled in its MavC-binding pocket in an almost identical fashion in all four structures reported here, in support of an effective increase in Ub affinity when it is presented in the format of a unit tethered to Ube2N. Additionally, having both the acyl acceptor (Lys92$^{Ube2N}$) and acyl donor (Gln40$^{Ub}$) units in one tethered molecule permits efficient capture of the Ub-thioester intermediate through a transamidation reaction rather than allowing a futile reaction via hydrolysis (deamidation). Even using the disulfide substrate as a means to provide Ub to MavC at higher affinity, we found no detectable deamidation of Ub, whereas CHBP can efficiently deamidate Ub in the same substrate (Supplementary Fig. 2k). Altogether, the results indicate that the acquisition of the insertion domain by MavC and its evolution into a Ube2N-binding motif has shifted the balance in favor of the transglutamination reaction at the cost of Ub deamidation.

The structures of the substrate complexes along with that of the product-bound enzyme captured here provide striking details of the ubiquitination reaction from crystallographic snapshots indicating conformational changes along the reaction coordinate. The insertion domain appears to behave as an independent Ube2N-binding domain but when built into MavC its rotational states provide specific functionality during the transamidation reaction. We propose the following model to capture the essential features of substrate recognition and transglutaminase-mediated ubiquitination catalyzed by MavC (Fig. 7a). Initial substrate engagement by the enzyme involves recruitment of the E2 component of a conjugate through interactions primarily involving the insertion domain. This is accompanied by rotation of the insertion domain to bring the E2 into an approximate productive arrangement and allowing placement of Ub into its binding pocket, as captured in two of the structures of the substrate complexes (C222$_1$ and R3). A further pendulum-like rotation of the insertion domain maximizes interaction of the E2 with MavC by pulling the E2 closer toward the active site, as captured in the P6$_5$ substrate complex and in the product complex. In all the substrate-bound structures, the Ub subunit is ready to undergo the first step of the reaction—attack of the Cys74$^{MavC}$ at Gln40$^{Ub}$. The resulting thioester intermediate is protected by the distinctly hydrophobic nature of the MavC active site pocket (Tyr254 and Trp255), which faces the 3$_{10}$-helix of Ube2N carrying the key Lys. Unfolding of this helix brings the Lys residue into an attacking position in the active site, as captured in the product-bound complex.

The structurally related *L. pneumophila* effector MvcA has recently been found to use a broadly similar interface, driven by the insertion domain, to bind Ube2N. However, it catalyzes the reverse reaction of MavC, removing Ub from Ube2N through hydrolysis and thereby serving to regulate MavC's activity[26]. Further work will be required to identify the basis of this key difference.

While MavC shares the Ub-binding site and core fold of the Cif family, our results show that it has diverged both structurally and functionally, having effectively lost the original function of Ub

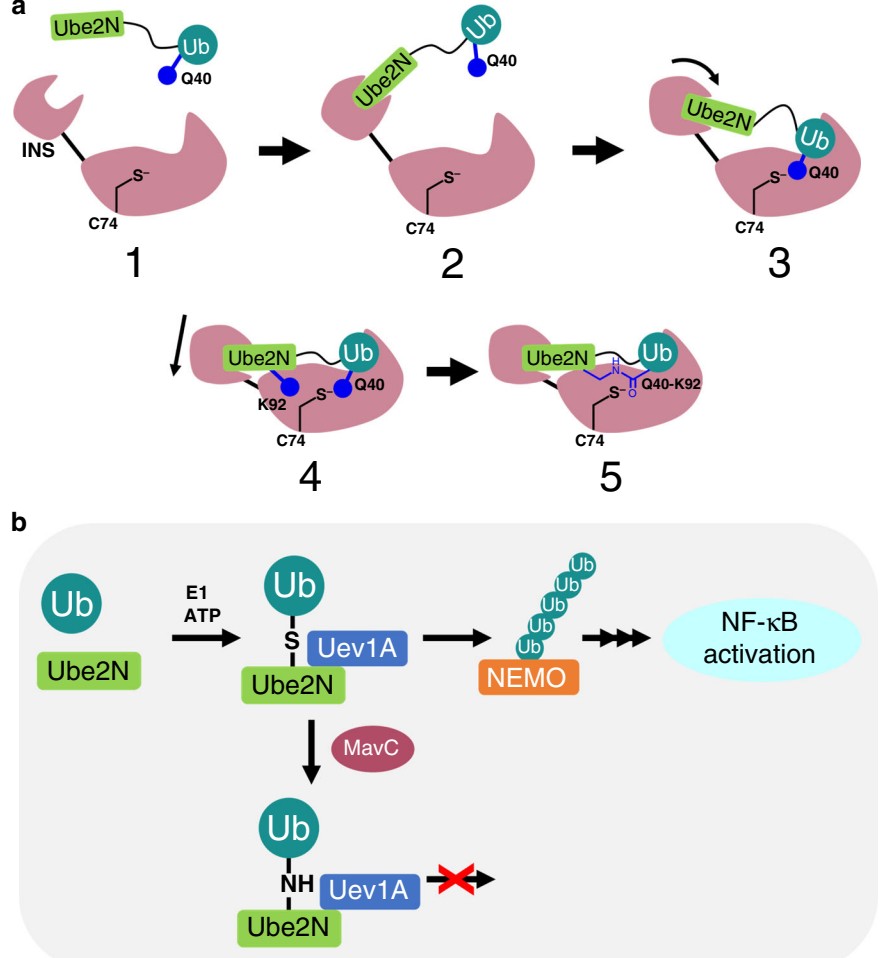

**Fig. 7 Overall scheme of MavC-catalyzed ubiquitination. a** Cartoon representation of Ube2N–Ub binding to MavC (1). INS domain of initially captures Ube2N moiety of substrate (2), moves to a rotated position where Ub has bound in the active site (3), and further adjusts to position Ube2N for catalysis (4). **b** Scheme of MavC ubiquitination and the consequences thereof. K63 chain formation is prevented by ubiquitination of Ube2N, resulting in a dampening of the immune response.

and NEDD8 deamidation through a lower Ub affinity. Instead, it preferentially attacks the Ube2N–Ub conjugate to turn off Ube2N's ability to generate K63 poly-Ub chains. Despite both free Ub/NEDD8 deamidation and Ube2N ubiquitination ultimately being shown to lead to decrease in NF-κB activation, we speculate that the divergence of MavC occurred to accommodate other *L. pneumophila* effectors that utilize the host's free Ub such as the E3 ligases LegU1 and SidC[36,37] and the noncanonical ligases of the SidE family. Poisoning of the host cell's supply of Ub by deamidation could antagonize these other effectors' activity. Therefore, MavC may satisfy a need for an alternative method of attenuating host immune signaling without perturbing the free Ub pool. How hampering the host's ability to make K63 poly-Ub chains may lead to other cellular effects remains to be elucidated.

## Methods

**Cloning, expression, and purification of recombinant proteins.** MavC$_{1–384}$ was PCR amplified from a plasmid encoding full-length MavC or MavC C74A using a PCR premix kit (Bioneer) cloned into pGEX-6p-1 plasmid (GE Healthcare). The resulting expression plasmid was transformed into BL21(DE3) strain of *E. coli* (Novagen). Cells were grown in LB media at 37 °C to an OD$_{600}$ of 0.6, cooled to 18 °C, and induced overnight at the reduced temperature by addition of 0.35 mM isopropyl-1-thio-β-D-galactopyranoside (IPTG). Cells obtained from 6 L bacterial culture were resuspended in phosphate-buffered saline (PBS) buffer (pH 7.4) supplemented with 0.4 M KCl (GST binding buffer). Resuspended cells were

disrupted under high pressure using a French press. The lysate was centrifuged for an hour at 100,000 × g at 4 °C, the supernatant was passed through a self-packed column of 5 mL glutathione-Sepharose resin (GE Healthcare) pre-equilibrated with GST-binding buffer. Following this, the resin was washed with five column volumes of GST-binding buffer. The bound fusion proteins were eluted with GST elution buffer (250 mM Tris pH 8, 500 mM KCl, and 10 mM reduced glutathione). The protein eluted off the column was dialyzed against two changes (4 L) of dialysis buffer (1× PBS supplemented with 0.5 M KCl and 1 mM DTT) at 4 °C to remove excess glutathione while being incubated with GST-tagged PreScission Protease as per the manufacturer's recommendation (GE Healthcare). The dialyzed sample was again passed through 5 mL of pre-equilibrated GST column to capture both free GST tag and the GST-tagged protease. The pure GST tag-cleaved protein, collected in the flow through, was concentrated to 44 mg/mL and buffer-exchanged into protein storage buffer (50 mM Tris pH 7.4, 50 mM NaCl and 1 mM DTT). The purity and the homogeneity of protein samples from every stage of expression and purification were monitored by sodium dodecyl sulfate polyacrylamide gel electrophoresis (SDS-PAGE). MavC-INS (residues 128–225), MavC$_{1–462}$, MavC$_{8–400}$, all of the MavC mutants used in this study, E1, Ubc13, Ube2R1, Ube2S, and Uev1A were also expressed and purified similarly. E1, Ubc13, Ube2R1, and Ube2S constructs were obtained from Genentech, and Uev1A was obtained from Yusuke Sato (University of Tokyo).

Ube2N (full-length) cloned into pET-SUMO was purchased from Addgene (Plasmid #51131). The plasmid, transformed into BL21(DE3) strain of *E. coli*, was expressed, lysed, and centrifuged same as the MavC constructs. The supernatant was passed through a pre-packed 5 mL HisTrap column (GE Healthcare) pre-equilibrated with binding buffer. Once the supernatant was loaded the resin was washed with five column volumes of binding buffer to wash off any unbound protein. This was followed by a 50 mM imidazole wash (binding buffer supplemented with 50 mM imidazole). The bound fusion protein was eluted with

elution buffer (1× PBS supplemented with 300 mM imidazole). The protein eluted off the HisTrap column was dialyzed against two changes (4 L) of dialysis buffer (1× PBS supplemented with 0.5 M KCl) at 4 °C to remove excess imidazole while being incubated with His-tagged SUMO protease (SENP2, purified via Ni$^{2+}$-affinity chromatography method described herein). The dialyzed sample was again passed through a pre-equilibrated 5 mL HisTrap column to capture both the tag and the protease. The tag-cleaved protein, collected in the flow through, was concentrated to 9.3 mM and buffer-exchanged into protein storage buffer (50 mM Tris pH 7.4, 50 mM NaCl, and 1 mM DTT).

Cif from *Yersinia pseudotuberculosis* was cloned into pQE30 vector, expressed, and purified similarly to Ube2N, with the exception of the tag cleavage step.

Ub constructs were cloned into the pRSET-A vector and transformed into the BL21(DE3) strain of *E. coli*. Protein expression was carried out as above. After cell disruption, lysates were subjected to heating at 70–80 °C for 15 min before ultracentrifugation, carried out as described above. Supernatant was adjusted to pH 4.5 with buffer containing 50 mM sodium acetate and spun down to remove precipitates, if necessary. The cleared supernatant was loaded onto a self-packed column of SP Sepharose Fast Flow resin (GE Healthcare) and eluted with a gradient of NaCl, up to 1 M. Fractions containing pure Ub, confirmed by SDS-PAGE analysis, were pooled, concentrated, and exchanged into protein storage buffer (50 mM Tris pH 7.4, 50 mM NaCl, and 1 mM DTT). The purity and the homogeneity of protein samples from every stage of expression and purification were monitored by SDS-PAGE.

Ube2N-SS-Ub, the disulfide conjugate, was prepared using a chemical method described by Lorenz et al.[38]. To prepare the conjugate, C-terminal Gly76$^{Ub}$ was first mutated to a cysteine by means of site-directed mutagenesis. The sequence-verified mutant was expressed and purified the same as the wild-type protein. The purified mutant, G76C$^{Ub}$, was first activated by mixing it with DTNB (5,5′-dithio-bis-nitrobenzoic acid) to give a final concentration of 250 μM G76C$^{Ub}$ and 2 mM DTNB in 0.1 M sodium phosphate (pH 8). The reaction was left to incubate overnight at 4 °C to form thiol-nitrobenzene labeled Ub-mutant, TNB-G76C$^{Ub}$. Excess DTNB was removed by buffer-exchanging the protein into assay buffer (50 mM Tris, 100 mM NaCl at pH 8). The disulfide conjugate was prepared by mixing equimolar ratio of Ube2N and TNB-G76C$^{Ub}$ in the assay buffer. The reaction was left to incubate for 3 h at 25 °C (with gentle rocking). The disulfide-linked protein complex, Ube2N-SS-Ub, was isolated by size-exclusion chromatography (Superdex 75) in the assay buffer. Fractions containing pure conjugate were pooled and concentrated to ~7 mg/mL.

To generate the ubiquitinated Ube2N product, purified MavC$_{1-462}$ was combined with Ube2N and Ub at a final concentration of 1 μM MavC, 25 μM Ube2N, and 100 μM Ub in a final volume of 30 mL. Reaction was allowed to proceed at 37 °C for 3 h in 50 mM Tris pH 7.4, 100 mM NaCl and 1 mM DTT, then concentrated to 1 mL, and passed through two Superdex 200 Increase 10/300 GL columns (GE Healthcare), connected in tandem, in the reaction buffer described above. Fractions containing pure Ub–Ube2N were pooled and concentrated to approximately 10 mg/mL.

Primers used for cloning the constructs in this study are listed in Supplementary Table 1.

**Complex formation, crystallization, data processing**. Complex of C74A-MavC$_{1-384}$ with Ube2N-SS-Ub was formed by mixing the two in a 1:1 molar ratio to give a final concentration of 32 mg/mL and incubating the mixture on ice for an hour. Crystals were grown by hanging drop vapor diffusion method at 21 °C. Crystals appeared in several conditions of the PEG-Ion screen from Hampton Research. The conditions identified from the commercial screens were optimized by reducing the protein concentration to 24 mg/mL during manual crystallizations. Data were collected from crystals grown in three different conditions. Crystal form I, grown in 0.1 M sodium acetate (pH 4.6) and 3.5 M sodium formate, diffracted to 1.97 Å at the Advanced Photon Source (APS) at Argonne National Laboratories on the LS-CAT 21-ID-G ($\lambda = 0.97$ Å) beam line. Data were processed and scaled using HKL2000 (ref.[39]) in the R3 space group with hexagonal setting, R3:H (Supplementary Table 2). Crystal form II, grown in 0.2 M potassium bromide and 30% PEG 2000 MME, diffracted to 2.3 Å at the Advanced Photon Source (APS) at Argonne National Laboratories on the LS-CAT 21-ID-G ($\lambda = 0.97$ Å) beam line. Data were processed and scaled using HKL2000 (ref.[39]) in centered orthorhombic space group, C222$_1$ (Supplementary Table 2). Crystal form III, grown in 0.2 M sodium formate and 20% PEG 3350, diffracted to 2.8 Å at the Advanced Photon Source (APS) at Argonne National Laboratories on the LS-CAT 21-ID-G ($\lambda = 0.97$ Å) beam line. Data were processed and scaled using HKL2000 (ref.[39]) in hexagonal space group, P6$_5$ (Supplementary Table 2).

Purified MavC$_{1-384}$ and Ub–Ube2N were mixed in a 1:1 molar ratio and incubated on ice for up to an hour. The resulting complex was concentrated to around 28 mg/mL. Crystals were grown by hanging drop vapor diffusion method at 21 °C in crystallization buffer containing 25% PEG 3350 and 0.2 M sodium malonate at pH 7.0. To confirm the presence of the desired complex in the crystals, SDS-PAGE analysis was performed after washing crystals with crystallization buffer and dissolving in water. The best diffracting crystals grew in an optimized crystallization condition containing 25% PEG 3350 and 0.2 M sodium malonate at pH 9.0 with 10 mM NiCl$_2$. A complete dataset to 2.07 Å was collected from a single crystal at the Advanced Photon Source (APS) at Argonne National Laboratories on

the LS-CAT 21-ID-G ($\lambda = 0.97857$) beam line. Data were processed and scaled using HKL2000 (ref.[39]) in the hexagonal space group, P6$_5$ (Supplementary Table 2).

MavC$_{INS}$ was crystallized using a solution containing a 1:2 molar ratio (14 mg/mL) of MavC$_{INS}$ to Ube2N and incubated at overnight 4 °C. Crystals were grown by hanging drop diffusion method at 21 °C. Best diffracting crystals were obtained by streak seeding in a buffer containing 2 M ammonium sulfate, 0.2 M lithium sulfate, and 0.1 M CAPS:NaOH; pH 10.5. A complete dataset at 1.52 Å was collected from a single crystal at the Advanced Photon Source (APS) at Argonne National Laboratories on the LS-CAT 21-ID-F ($\lambda = 0.978772$) beam line. Data were processed and scaled using HKL2000 (ref.[39]) in the trigonal space group, P3$_1$ (Supplementary Table 2).

**Structure determination and refinement**. The structures of MavC$_{1-384}$–Ube2N-SS-Ub complex in three different space groups was determined by maximum likelihood molecular replacement using the program PHASER[40] from CCP4 suite[41]. The initial search models used were native MavC$_{7-384}$ (PDB code 5TSC[21]), Ube2N (PDB code 2C2V[42]), and Ub (PDB code 1UBQ[43]). The asymmetric unit consists of one complex in each of the three space groups. Iterative rounds of model building with the program COOT[44] and refinement with the program PHENIX[45] was used to arrive at the final structures that were validated through MolProbity[46] and deposited in the Protein Data Bank.

The structure of the complex between MavC$_{1-384}$ and Ube2N-SS-Ub in the P6$_5$ space group (early complex), determined at 2.80 Å resolution (Supplementary Table 2), has one molecule of MavC$_{1-384}$ engaging with one molecule of Ub–Ube2N in the asymmetric unit. Analysis of the Ramachandran plot[47] indicated that 95.3% of residues fall in the most favored region and 4.7% in the additional allowed regions of the plot, none in the disallowed region of the plot. Electron density for most of the residues from MavC, Ub, and Ube2N are well resolved in the crystal structure. The final structure was validated with MolProbity[46] and deposited in the Protein Data Bank (PDB code for the coordinate and the reflection files is 6UMP).

The structure of the complex between MavC$_{1-384}$ and Ube2N-SS-Ub in the R3 space group (attacking complex I), determined at 1.97 Å resolution (Supplementary Table 2), has one molecule of MavC$_{1-384}$ engaging with one molecule of Ube2N-SS-Ub in the asymmetric unit. Analysis of the Ramachandran plot[47] indicated that 98.8% of residues fall in the most favored region and 1.2% in the additional allowed regions of the plot, none in the disallowed region of the plot. Electron density for almost all of the residues from MavC, Ub, and Ube2N are well resolved. Residues from 89 TO 94 in Ube2N have almost no density and hence have not been modeled in the structure. A total of 298 water molecules (an average B-factor of 42.6 Å$^2$) were observed in the asymmetric unit (PDB code for the coordinate and the reflection files is 6ULH).

The structure of the complex between MavC$_{1-384}$ and Ube2N-SS-Ub in the C222$_1$ space group (attacking complex II), determined at 2.34 Å resolution (Supplementary Table 2), has one molecule of MavC$_{1-384}$ engaging with one molecule of Ube2N-SS-Ub in the asymmetric unit. Analysis of the Ramachandran plot[47] indicated that 97.8% of residues fall in the most favored region and 2.2% in the additional allowed regions of the plot, none in the disallowed region of the plot. Electron density for almost all of the residues from MavC, Ub, and Ube2N are well resolved. Residues from 89 to 94 in Ube2N have almost no density in this complex as well and hence have not been modeled in the structure. A total of 168 water molecules were observed in the asymmetric unit (PDB code for the coordinate and the reflection files is 6UMS).

The structure of MavC$_{1-384}$–Ub-Ube2N complex (product complex) was determined by maximum likelihood molecular replacement using the program PHASER[40] from CCP4 suite[41]. The initial search models used were native MavC$_{7-384}$ (PDB code 5TSC[21]), Ube2N (PDB code 2C2V[42]), and Ub (PDB code 1UBQ[43]). The asymmetric unit consists of one complex. Iterative rounds of model building with the program COOT[44] and refinement with the program PHENIX[45] resulted in the final structure for all data between 40 and 2.1 Å resolution (Supplementary Table 2). The final structure was validated through MolProbity[46] and deposited in the Protein Data Bank (PDB code for the coordinate and the reflection files is 6P5B). The structure of the complex between MavC$_{1-384}$ and ubiquitinated Ub (Ub–Ube2N), determined at 2.07 Å resolution (Supplementary Table 2), has one molecule of MavC$_{1-384}$ engaging with one molecule of Ub–Ube2N in the asymmetric unit. Analysis of the Ramachandran plot[47] indicated that 97.9% of residues fall in the most favored region and 2.1% in the additional allowed regions of the plot, none in the disallowed region of the plot. Electron density for the main-chain of the MavC$_{1-384}$ construct along with four non-native residues at the N-terminal end remaining after the cleavage of the GST affinity tag are well resolved in the crystal structure. A total of 197 water molecules (an average B-factor of 43.4 Å$^2$) were observed in the asymmetric unit. A separate model with the catalytic Cys of MavC included was generated using COOT.

The MavC insertion domain (MavC$_{INS}$) structure was determined by maximum likelihood molecular replacement using the program PHASER[40] in the PHENIX[45] suite. The initial search model used was the insertion domain from MavC$_{7-384}$ (PDB code: 5TSC[21]). The asymmetric unit contained two molecules, which were consistent with the formation of a crystallographic dimer. The structure was taken through multiple steps of model building with the program COOT[44] and

refinement with the program PHENIX[45] which resulted in a final structure for all data between 29.8 and 1.53 Å resolution (Supplementary Table 2). The final structure was validated through MolProbity[46] and deposited with the Protein Data Bank (PDB code for the coordinates and structure factors is 6P5H).

Residues where poor electron density was observed were modeled as either alanines or glycines. Interface area and residues at the interface in all the complexes presented in this study were computed using the web-based server, PISA, at the European Bioinformatics Institute (http://www.ebi.ac.uk/pdbe/prot_int/pistart.html)[48]. All structures in this work were rendered and presented using Pymol (http://pymol.org).

**Mutagenesis**. Plasmids harboring the desired mutations and/or truncations of MavC were constructed via site-directed mutagenesis using mutagenic primer pairs. The resulting mutant PCR products were digested with *Dpn*I to remove the methylated templates and transformed into *E. coli* DH5α (home-made competent cells). Presence of desired mutations was confirmed by sequencing before being transformed into *E. coli* BL21(DE3) for protein expression and subsequent purification, similar to the aforementioned MavC constructs.

Primers used for mutagenesis in this study are listed in Supplementary Table 1.

**GST pulldown assays**. The pulldown assays were performed with GST-fusion proteins of MavC$_{1-462}$, MavC$_{INS}$, and MavC$_{\Delta INS}$ as probe proteins to pulldown Ube2N. A 100 μL of a 50% slurry of glutathione-agarose beads were equilibrated with 10× bed volume of 1× PBS buffer; pH 7.4. The beads were then centrifuged for 5 min at 5000 rpm and the supernatant discarded. This wash step was repeated twice more. A total of 100 μL of a 100 μM stock of the probe protein(s) was used to charge the beads. These were then incubated at 4 °C with end-over-end mixing for 4 h, to ensure that the bait protein bound to the beads. The loaded beads were centrifuged for 5 min at 5000 rpm and the flow through collected. The beads were washed as before and the wash collected after each step. Following the washes, the charged beads were incubated with 100 μL of a 100 μM stock of Ube2N. Binding was allowed to proceed overnight at 4 °C with end-over-end mixing. After incubation, the beads were centrifuged and washed as above, again collecting the flow through and wash at every step. Proteins were eluted by incubating the beads in 50 μL of GST elution buffer (250 mM Tris pH 8, 500 mM KCl, and 10 mM reduced glutathione) for 15 min at room temperature followed by centrifugation. The elution step was repeated once more. The collected samples were analyzed by SDS-PAGE.

**Biolayer interferometry**. Inactive (C74A) poly-His-tagged MavC constructs for use in the BLI studies were generated using two-step PCR (Megaprimer method) and verified using DNA sequencing. Primers used for cloning the constructs in this study are listed in Supplementary Table 1. The His-tagged proteins were expressed and purified as described above for Ube2N. Homogeneity of all the purified proteins was confirmed by SDS-PAGE analysis. The MavC constructs used for these studies were diluted in BLI buffer (1× PBS containing 0.05% v/v Tween-20 and 0.1% w/v BSA) to a concentration of 25 μg/mL. The analytes were also diluted in the same buffer at the following concentrations: Ub (2 mM), Ube2N (100 μM), Ub-SS-Ube2N (100 μM), and Ub–Ube2N (100 μM). Serial dilutions of each analyte were prepared, in replicates of three, for analysis. One Ni-NTA biosensor was used for each $K_D$ measurement, dipping the MavC protein loaded tip into wells that contained an analyte, starting with the lowest concentration of analyte first. The direct binding experiment was performed for 120 s for association and 100 s for dissociation in BLI buffer using a biolayer interferometer (BLI), Octet Red 96 system (Pall ForteBio, Corp., Menlo Park, CA, USA), and data acquired using the ForteBio Data Acquisition 8.2 software (Pall ForteBio Corp., Menlo Park, CA, USA). Association responses (from 110 to 115 s) was averaged and plotted in BAL Octet Data Analysis Software. The data were fit to a non-linear regression one site—specific binding model to determine the $K_D$.

**Protein expression, purification, and NMR spectroscopy**. $^{15}$N-labeled Ub and Ube2N were grown in minimal MOPS media supplemented with $^{15}$NH$_4$Cl. $^2$H,$^{15}$N-labeled Ub$_{WT}$, Ub$_{G76C}$, and Ube2N$_{WT}$ were grown in minimal M9 media prepared in D$_2$O and supplemented with $^{15}$NH$_4$Cl. In all cases, Ub and Ube2N constructs were expressed and purified as previously described[31]. The disulfide Ube2N-SS-Ub conjugate was prepared as described above[38]. NMR data were collected at 25 °C on a 500Mhz Bruker AVANCE III, or 600 MHz AVANCEII fitted with a cryoprobe. NMR samples were prepared in 25 mM Tris, 100 mM NaCl, pH 7.6 with the addition of 5% D$_2$O. Samples typically contained 150 mM isotopically labeled protein and unlabeled WT or C74A-MavC$_{1-384}$ at concentrations from 0 to 225 mM. Data were processed with NMRPipe[49] and analyzed using NMRViewJ[50]. The magnitude of chemical shift changes for specific resonances followed during NMR titration experiments were determined using the following equation:

$$\delta = \left( \left[ \delta_H^2 + a\delta_N^2 \right] / 2 \right)^{1/2} \tag{1}$$

where $\delta_H$ and $\delta_N$ are the $^1$H and $^{15}$N chemical shift changes, respectively, and 0.14 was used for the scaling factor $a$[51].

**Ubiquitination and deamidation assays**. To analyze the ubiquitinating activity of MavC mutants, purified MavC$_{1-462}$ constructs (wild type or mutants) were combined with Ube2N-SS-Ub at a final concentration of 0.005 μM MavC and 25 μM Ube2N-SS-Ub, and incubated at 37 °C for 30 min in reaction buffer (50 mM Tris pH 7.4, 100 mM NaCl). The reaction products were analyzed by SDS-PAGE and visualized with Coomassie Blue. To analyze the quadruple mutants an compare them with other mutants, an increased incubation time of 60 min and a [MavC] of 0.05 μM was utilized. On the other hand, to analyze the ubiquitinating activity of MavC$_{\Delta INS}$ and also the ubiquitinating activity of MavC against Ubc13, UbE2S, and UbE2R1, 0.5 μM MavC, 25 μM Ube2N, and an extended incubation time of 60 min was used, and the Ub concentration was increased to 100 μM in an attempt to promote ubiquitination.

Ub deamidating assays were performed by combining purified MavC$_{1-462}$ constructs (wild type or mutants) with Ub at a final concentration of 0.5 μM enzyme and 100 μM Ub. The reactions were incubated at 37 °C for 30 min in reaction buffer (50 mM Tris pH 7.4, 100 mM NaCl, 1 mM DTT). The deamidation reaction products were analyzed by Native-PAGE and visualized with Coomassie Blue.

MavC's deamidating activity on the disulfide conjugate was tested by combining purified MavC$_{1-462}$ with Ube2N-SS-Ub at a final concentration of 0.005 μM MavC and 25 μM Ube2N-SS-Ub and incubated at 37 °C for 30 min in reaction buffer (50 mM Tris pH 7.4, 100 mM NaCl). The reaction products were analyzed by Native-PAGE and visualized with Coomassie Blue. As a control to observe the migration of deamidated Ube2N-SS-Ub, a reaction was run utilizing the known deamidase Cif at a concentration of 0.5 μM.

To determine the Michaelis–Menten kinetic parameters of the Ube2N ubiquitinating activity of MavC, reactions were conducted with MavC (0.5 μM), and varying concentrations of Ube2N at 37 °C for 30 min. Reactions were quenched with SDS-PAGE loading buffer and separated by SDS-PAGE along with Ube2N standards of known concentrations and visualized with Coomassie Blue. Gels were analyzed with ImageJ, and a standard curve was generated using the band intensities of the Ube2N standards. This standard curve was used to quantify Ube2N produced from each reaction. Data were fit to the Michaelis–Menten equation. Linear regression and plotting were performed using SigmaPlot. Reactions were performed in triplicate for kinetic analysis. Error bars were generated via the standard deviation.

To determine the Michaelis–Menten kinetic parameters of the Ub deamidating activity of MavC, reactions were conducted with MavC (0.5 μM), and varying concentrations of Ub at 37 °C for 30 min. Reactions along with Ub standards of known concentrations were separated by Native-PAGE. Gels were analyzed by ImageJ as described above. All reactions were performed in triplicate. Error bars were generated via the standard deviation.

To determine the activity of MavC against the Uev1a:Ube2N-SS-Ub complex, varying amounts of Uev1a were incubated with Ube2N-SS-Ub for 10 min prior to addition of 0.005 μM MavC. Reactions were performed at 37 °C for 30 min. Reactions were quenched with SDS-PAGE loading buffer, separated by SDS-PAGE, and visualized with Coomassie Blue.

**MavC-mediated ubiquitination of Ube2N during *L. pneumophila* infection**. *L. pneumophila* strains were grown to post-exponential phase (OD$_{600}$ = 3.2–3.8) in AYE medium at 37 °C and then induced for 4 h with 0.2 mM IPTG before infection. *L. pneumophila* strains were obtained from prior studies[22,52]. Raw264.7 cells or U937 cells were infected with *L. pneumophila* strains at an MOI of 10 for 2 h. Cells were washed three times with PBS and then lysed with 0.2% saponin on ice for 30 min. Cell lysates were resolved by SDS-PAGE and probed with MavC (1:5000 dilution)[22]-specific and Ube2N (1:1000 dilution)-specific antibodies. Anti-Ube2N antibody was purchased from Thermo Fisher Scientific (catalog # 37-1100). Tubulin and ICDH were used as a loading controls and probed using anti-tubulin antibody (1:10,000 dilution) from DSHB (catalog # E7) and anti-ICDH (1:10,000)[53].

**E1 charging assay**. To compare the ability of E1 to charge Ube2N versus Ub–Ube2N, a reaction mixture of 0.5 μM E1 enzyme, 200 μM Ube2N or Ub–Ube2N, 400 μM Ub was conducted in a reaction buffer consisting of 50 mM Tris pH 7.4, 100 mM NaCl, 2.5 mM ATP, 5 mM MgCl$_2$. Reactions were allowed to proceed for 30 min at 37 °C before quenching with either reducing or non-reducing SDS-PAGE buffer, separated by SDS-PAGE and visualized with Coomassie Blue.

**NF-kB activation assay**. HEK293T cells were grown to 70% confluence in 24-well plates, and transfected with 100 ng NF-κB reporter plasmids and 10 ng of plasmid that directs the expression of *Renilla* luciferase in pRL-SV40 (Promega) as internal control. Four hundred nanograms 4xFlag-*MavC* vector or its mutant and 400 ng 4xFlag-*TRAF6* were co-transfected at the same time. After 24 h, the cells were then collected and lysed for NF-κB luciferase reporter assay following the manufacturer's protocols (Promega cat. no. E1910). Briefly, cells were lysed with 100 μL passive lysis and 20 μL of cell lysates transferred to a 96-well plate. Dispense 100 μL Luciferase assay buffer containing Firefly luciferase substrate and measure Firefly luciferase activity. After that, 100 μL Luciferase Stop and Glo reagent was added and *Renilla* luciferase activity measured. The expression of MavC or its mutant was

probed in lysates of transfected cells and the blots shown are representatives of at least three independent experiments. Anti-MavC antibody[22] was used at a dilution of 1:5000. Tubulin was used as a loading control and probed using anti-tubulin antibody (1:10,000 dilution) from DSHB (catalog # E7).

**Statistical methods**. The gels presented in this study are representative of three different experiments performed independently, with similar results obtained (Figs. 2e, 4c, d, 5d, e, Supplementary Figs. 1d, 2k, 3a-c, e, 4c and 5a, b, e). No statistical method was used to predetermine sample size. The experiments were not randomized and were not performed with blinding to the conditions of the experiments.

**Reporting summary**. Further information on research design is available in the Nature Research Reporting Summary linked to this article.

## Data availability
Coordinates of all five structures have been deposited into the Protein Data Bank under accession codes 6UMP, 6ULH, 6UMS, 6P5B and 6P5H. The source data underlying Figs. 2a, b, e, 4c, d, 5d, f, and Supplementary Figs. 1e, 2a, b, 3a, c, 3e, 4b, c, and 5a, b are provided as a Source Data file. Other data are available from the corresponding authors upon reasonable request.

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

## Acknowledgements
This work was funded by National Institute of Health Grants R01GM126296 (to C.D.), T32GM132024 (to S.K.), R01AI127465 (to Z-Q.L.), and R01 GM088055 (to R.E.K.). We thank our staff contacts, Monica Green and Joe Brunzelle, on the LS-CAT beamlines, 21-ID-F and 21-ID-G, at the Advanced Photon Source (Argonne National Laboratory) for their support during X-ray data collection. We thank Ronald Stenkamp for help comparing Ube2N structures. We acknowledge Genentech, Inc. for the gifts of the E1, Ubiquitin, Ube2R1, and Ube2S plasmids, and Yusuke Sato (University of Tokyo) for the Uev1A plasmid. The contents of this publication are solely the responsibility of the authors and do not necessarily represent the official views of NIH.

## Author contributions
K.P., S.I., Z-Q.L., P.S.B., R.E.K., and C.D. designed and interpreted experiments. P.S.B. and R.E.K. planned and carried out NMR analyses and designed biochemical assays. J.F. and Z-Q.L. performed *Legionella* infection and MavC transfection experiments. S.K. performed BLI analyses. K.I.N.T. performed pulldown experiments. K.P and S.I. performed all other experiments including crystallization and structure determination of the protein complexes. K.P., S.I., P.S.B., R.E.K., and C.D. wrote the paper with editorial input from all authors.

## Competing interests
The authors declare no competing interests.
