## [Peer Review File · Nature Communications]

Reviewers' comments:

Reviewer #1 (Remarks to the Author):

In this paper, Puvar K. et al. reported an interesting discovery that the Legionella effector MavC targets the activated, thioester-linked Ube2N~Ub conjugate as the real substrate, not free Ub or Ube2N as previous studies reported, to catalyze an intramolecular transglutamination reaction and covalently crosslink the Ube2N and Ub subunits for noncanonical ubiquitination. They presented a series of biochemical data and four complex structures of MavC with Ube2N and Ub. The structures clearly uncovered the molecular basis for recognition of the Ube2N~Ub conjugate and uncovered the catalytic mechanism with protein dynamics in the transglutamination reaction. The whole manuscript is well-written and should be published in Nature Communication after revision.

I have the following concerns:

- 1, In the Table S1 for structure determination, the I/sigma I value of the last shell in MavC1-384_Ub-SS-Ube2N were much less than 2. The authors should cut it to be more than 1.5 or 2 and re-refine the structure.
- 2, In Figure 4C, the authors should analyze the effects of the mutants of MavC on the transglutamination reaction, not on the deamidation reaction.
- 3, In Figure 5D, the authors did not show the effect of Tyr47 of MavC. This mutant should be included. In addition, the role of Tyr47 in substrate recognition or structural stability of MavC should be presented in a structural figure.
- 4, The data in Figure 5E is not consistent fully with those in Figure 5F. The YYE mutant still inhibits NF-kB signaling. The author should test the effects on NF-kB activation by TRAF6 overexpression.
- 5, All mutants of the Ub- and Ube2-binding residues of MavC (in Figures 4 and 5) should be test for their effects on the transglutamination reaction in vivo. The completely inactive mutants of the binding residues of MavC also need be tested during infection.
- 6, In Figure 7A, the author tested the effect of adding Uev1 on the transglutamination reaction. Uev1A did not significantly affected the modification. But the authors need to test the effect of the crosslinking of Ube2N-Ub on the interactions between Ube2N and Uev1A for further confirming their conclusion.

Reviewer #2 (Remarks to the Author):

In their manuscript, Puvar et al. explore the structural basis of MavC catalyzed ubiquitination of the Ub conjugating enzyme Ube2N. In two previous studies, MavC has been reported to act as a transglutaminase that catalyzes isopeptide bond formation between the Ub Q40 and Ube2N K92 side chains without requiring Ub activation by a canonical E1 enzyme or ATP. Instead, MavC deamidates Ub Q40 to form a thioester intermediate with the MavC catalytic Cys prior to reacting with the Ube2N K92 amine.

In this study, the authors identified the Ube2N~Ub thioester as primary MavC substrate. The authors determined crystal structures of a Ube2N~Ub G76C disulfide as a thioester mimic and the isopeptide linked Ube2N-Ub in complex with a catalytically incompetent MavC C74A mutant. The structures of the substrate and product bound states of MavC offer valuable snapshots of conformational changes in MavC and Ube2N along the reaction trajectory and are validated with biochemical assays.

The manuscript is an insightful contribution to our understanding of how bacterial effectors target Ub signaling pathways and elucidates how MavC favors Ub transglutamination over deamidation activity. Overall, the paper is based on solid experimental data that support the authors' conclusions. However, the findings reported in the last subsection of the manuscript seem somewhat preliminary. Moreover, the interpretation of the structures could be more detailed and

many figure legends should be considerably more explicit in order for this study to be suitable for publication in Nature Communications.

Major issues:

- The affinities of MavC to Ube2N alone and the Ube2N-Ub thioester mimic are virtually identical (Fig. 2a). However, the Ube2N-thioester linked Ub interacts with MavC in the crystal structures and in the NMR binding studies. The Ub binding surface should contribute to the K_d of the Ube2N-SS-Ub:MavC complex. How do the authors explain this apparent discrepancy?
- In previous studies, free Ub and even a Ub G75A/G76A mutant that cannot form a thioester intermediate with Ube2N were shown to be linked to Ube2N by MavC. The authors should discuss these results.
- The quality of the Ube2N-thioester linked Ub 15N spectra in Fig. 2d is quite poor and contains many signals with low intensities. Are these due to chemical exchange or due to impurities or degradation of the sample? It would be helpful if the authors showed an overlay of the 15N spectra of free Ub with the Ube2N-SS-linked Ub in the Supplement and labeled the Ub resonances. Do the CSPs in Fig. 2d agree with the Ub binding surfaces of the crystal structures?
- Fig. 2e: The high concentration of DTT (1 mM) in the reaction buffer compared to the Ube2N~Ub disulfide (25 μM) should reduce the disulfide to free Ube2N and Ub. The authors should show loading controls in the absence of MavC for each time point to demonstrate the integrity of the Ube2N~Ub disulfide during the reaction.
- Fig. 5e,f: Although the YYE mutant is virtually incapable of performing transglutamination, it still effectively down-regulates NF-κB compared to the C74A mutant. The authors should discuss this result. Fig. 5f: The MavC gel (six lanes) does not match with the Tubulin gel (seven lanes). What does the bar below the bar chart refer to? Significance should be tested in the bar chart and the gels should be labeled. The methods for measuring NF-κB activity should be described in more detail. The error bars and number of experiments are not described.
- The paper falls short of analyzing the conformation of the Ube2N~Ub thioester and Ube2N-Ub complex. The authors should compare / superimpose their structures with known structures of Ube2N (or Ubc13)~Ub conjugates. Would E3 binding clash with the MavC interaction? A more thorough analysis of the Ube2N-Ub isopeptide would be helpful to understand the mechanism underlying E2 inhibition and NF-κB inactivation. Does the structure of the product complex provide any cues in this direction? A CSP analysis of ¹⁵N Ub isopeptide linked to Ube2N might be useful in this respect.
- The subsection describing the role of Uev1a is somewhat preliminary. The authors should therefore move Fig. 7a-b to the Supplement and substantiate their speculations by adding a structural model of the Ube2N-Ub:MavC complex including Uev1a. The units in Fig. 7a for the time points are missing.
- Fig. S5 nicely summarizes the structural data. I would therefore recommend replacing Fig. 7a,b with Fig. S5. The Ube2N C87 and K92, Ub Q40 and MavC C74 side chains should be labeled. The isopeptide linked Ube2N-Ub / product complex should be shown as well. It would also be helpful to include these cartoons on top of the structures shown in Fig. 6a-c.

Minor issues:

- The formatting of Greek letters and superscripts is missing / has been lost at several positions. A single font type should be used for all figures.
- Throughout the text, the authors should make clear which catalytic Cys / thioester is referred to, Ube2N or MavC.

- Ube2N is also referred to as Ube2N and UBE2N. The spelling should be consistent throughout the paper. Similarly, a space should always be included between values and their units.

- Fig. S2: Fig. S2d: The ^{15}N axis should be labeled, since it should be different from Fig. S2c (HSQC vs. TROSY spectrum). Fig. S2e and f: The corresponding NMR spectra should be shown. What is the CSP cut-off used in Fig. 2f? Fig. S2h: The units for the time points are missing.

- Fig. S3: The numbering of the panels is not consistent with the text, e.g. Fig. S3e is mentioned in the text before Fig. S3a. Fig. S3a: The YYE mutant should be spelled out in the same way as the other mutants. Fig. S3c: The units for the time points are missing. Fig. S3e: The gel is not described in the legend.

- Fig. 4c: The figure legend should be more explicit. How are the Ub species visualized? What does the "Ctrl" lane refer to?

- Fig. S4d: The figure legend should be more explicit. "corresponds to maximum rmsd" of what?

- Constructs and purification of E2 enzymes other than Ube2N and of Cif are not described.

- Software used for multiple sequence alignments is not described.

Reviewer #1 (Remarks to the Author):

In this paper, Puvar K. et al. reported an interesting discovery that the Legionella effector MavC targets the activated, thioester-linked Ube2N~Ub conjugate as the real substrate, not free Ub or Ube2N as previous studies reported, to catalyze an intramolecular transglutamination reaction and covalently crosslink the Ube2N and Ub subunits for noncanonical ubiquitination. They presented a series of biochemical data and four complex structures of MavC with Ube2N and Ub. The structures clearly uncovered the molecular basis for recognition of the Ube2N~Ub conjugate and uncovered the catalytic mechanism with protein dynamics in the transglutamination reaction. The whole manuscript is well-written and should be published in Nature Communication after revision.

I have the following concerns:

1, In the Table S1 for structure determination, the $I/\sigma I$ value of the last shell in MavC1-384_Ub-SS-Ube2N were much less than 2. The authors should cut it to be more than 1.5 or 2 and re-refine the structure.

We appreciate this concern. In our original submission we had used the $CC \frac{1}{2}$ metric as our high-resolution cutoff, as recommended by Karplus and Diederichs (Curr Opin. Struct. Biol. 2015). In responding to the reviewer's suggestion, we re-scaled the data for our two structures where the $I/\sigma I$ value was under 1.5, and re-refined them. The re-refined structures did not improve the final overall statistics beyond what had already been reported. Instead, they produced lower crystallographic and free-R factors. Thus, it appears that by rejecting some reflections by using the suggested cutoff criteria, we are compromising the quality of the originally reported model. Therefore, we would like to retain our original cutoff criteria.

2, In Figure 4C, the authors should analyze the effects of the mutants of MavC on the transglutamination reaction, not on the deamidation reaction.

Thank you for this feedback, we have performed this analysis and found that transglutamination is also impaired when Ub-binding mutations of MavC are present, consistent with the structural data. We have therefore added this figure in Figure 4 between the original Figure 4C and 4D.

3, In Figure 5D, the authors did not show the effect of Tyr47 of MavC. This mutant should be included. In addition, the role of Tyr47 in substrate recognition or structural stability of MavC should be presented in a structural figure.

We have updated Figure 5C to show the interaction of Tyr47 of MavC with Ube2N. When this residue is mutated to Ala, transglutaminating activity of MavC is impaired (Please see Figure S3).

4, The data in Figure 5E is not consistent fully with those in Figure 5F. The YYE mutant still

inhibits NF- κ B signaling. The author should test the effects on NF- κ B activation by TRAF6 overexpression.

Thank you for the suggestion. Figures 5E and 5F have been replaced with new results that address these concerns; specifically, we have conducted the NF- κ B activation assay with the triple mutant and two new quadruple mutants (please see response to point 5 below) in the presence of TRAF6 overexpression, as suggested by the reviewer. The quadruple mutants displayed further defects in the transglutaminase catalysis compared to the triple mutant, as revealed under longer reaction times where the triple mutant does seem to show appreciable activity. The results show clearly that disruption of Ube2N binding of MavC can substantially restore NF- κ B activation under TRAF6 overexpression conditions. These more drastic quadruple mutants do exhibit stronger restoration of NF- κ B response when stimulated with TNF- α . In light of the reviewer's suggestion we decided to include the new TRAF6 overexpression results in Fig. 5 in place of the results from TNF- α stimulation conditions. Overall, the new results are indeed consistent with our mutational analysis looking at transglutaminase activity under biochemical assay conditions (Figure 5E, F, S3b, c) A discussion of these new results has now been included in the text.

5, All mutants of the Ub- and Ube2-binding residues of MavC (in Figures 4 and 5) should be tested for their effects on the transglutamination reaction *in vivo*. The completely inactive mutants of the binding residues of MavC also need to be tested during infection.

We do appreciate the suggestion. However, in our biochemical assays, we find that individual point mutants do not show a substantial defect in transglutamination under conditions of longer reaction times than shown in Figure 5D. Therefore, we believe that when translated to cellular-based assays (which have much longer time scales), these point mutants may not produce unambiguous results. Instead, we decided to create new mutants that would produce more severe defects in biochemical assays and examined them under infection conditions. These mutants were defective in our biochemical transglutaminase assay even under conditions of longer reaction time (Figure S3c). The same mutants exhibit a much more pronounced defect in the Ube2N ubiquitination under infection conditions, apparently at a similar magnitude to the catalytically inactive C74A mutant (Figure 5e).

We have now changed the passage in the text describing our biological experiments as follows:

“We also created more mutants by additionally changing either Met317 or Trp255 to alanine in the YYE mutant (YYE/M317A, YYE/W255A). Unlike the single point mutants or YYE, which show appreciable activity under longer reaction times in biochemical assays, these quadruple mutants were much more defective (**Fig. S3b,c**). All of these mutants were translocated into infected cells at levels comparable to that of the wild type complementation. We then examined the levels of Ube2N ubiquitination in infected cells. Also, we examined the ability of MavC and the mutants to attenuate NF- κ B activation under conditions of TRAF6 overexpression. Whereas the activity of the YYE mutant has almost completely lost the ability to modify Ube2N and

displays defects in attenuating NF- κ B activation, the quadruple mutants were further impaired to levels comparable to the catalytically inactive C74A mutant, in line with the biochemical activity assay results (Fig. 5e, f. S3c).”

6, In Figure 7A, the author tested the effect of adding Uev1 on the transglutamination reaction. Uev1A did not significantly affected the modification. But the authors need to test the effect of the crosslinking of Ube2N-Ub on the interactions between Ube2N and Uev1A for further confirming their conclusion.

We appreciate the feedback. Indeed we do observe co-elution of the disulfide crosslinked Ube2N-SS-Ub with Uev1A when they are combined and subjected to size-exclusion chromatography (Fig. S5A). Further, upon examination of the crystal structure of Ubc13~Ub-Mms2 complex reported by Eddins et al. (PDB#2GMI, Nat. Struct. Mol. Biol. 2006), the binding interface between Ubc13 and Mms2 (respective yeast analogs of Ube2N and Uev1A) appears to be some distance away from the Ube2N~Ub linkage. This structural analysis along with our experimental result suggests that the disulfide crosslink between C87 of Ube2N and G76 of Ub does not prevent Uev1A from binding. The modeled structure has been included as Figure S5C.

We have updated the text to reflect that Ube2N-Uev1A binding is not affected:

“Accordingly, we demonstrated that Uev1A can bind to Ube2N-SS-Ub as deduced from co-elution of the two proteins during size-exclusion chromatography (Fig. S5a)”

Reviewer #2 (Remarks to the Author):

In their manuscript, Puvar et al. explore the structural basis of MavC catalyzed ubiquitination of the Ub conjugating enzyme Ube2N. In two previous studies, MavC has been reported to act as a transglutaminase that catalyzes isopeptide bond formation between the Ub Q40 and Ube2N K92 side chains without requiring Ub activation by a canonical E1 enzyme or ATP. Instead, MavC deamidates Ub Q40 to form a thioester intermediate with the MavC catalytic Cys prior to reacting with the Ube2N K92 amine.

In this study, the authors identified the Ube2N~Ub thioester as primary MavC substrate. The authors determined crystal structures of a Ube2N~Ub G76C disulfide as a thioester mimic and the isopeptide linked Ube2N-Ub in complex with a catalytically incompetent MavC C74A mutant. The structures of the substrate and product bound states of MavC offer valuable snapshots of conformational changes in MavC and Ube2N along the reaction trajectory and are validated with biochemical assays.

The manuscript is an insightful contribution to our understanding of how bacterial effectors target Ub signaling pathways and elucidates how MavC favors Ub transglutamination over

deamidation activity. Overall, the paper is based on solid experimental data that support the authors' conclusions. However, the findings reported in the last subsection of the manuscript seem somewhat preliminary. Moreover, the interpretation of the structures could be more detailed and many figure legends should be considerably more explicit in order for this study to be suitable for publication in Nature Communications.

Major issues:

- The affinities of MavC to Ube2N alone and the Ube2N-Ub thioester mimic are virtually identical (Fig. 2a). However, the Ube2N-thioester linked Ub interacts with MavC in the crystal structures and in the NMR binding studies. The Ub binding surface should contribute to the K_d of the Ube2N-SS-Ub:MavC complex. How do the authors explain this apparent discrepancy?

Thanks for pointing this out. Indeed, from the crystal structure one may be led to believe that Ub ought to contribute to the K_d of Ube2N-SS-Ub. However, the structural data of the apo MavC shows that the insertion domain is sterically occluding Ub from binding. This suggests that Ub binding when tethered to Ube2N does not occur to a preorganized binding patch on MavC, rather is a result of movement of the insertion domain. We believe that this may explain the apparent absence of contribution of Ub to the binding affinity of Ub-SS-Ube2N for MavC. The binding appears to be dominated by interactions with the E2 part. In line with notion, a subset of Ub residues appear to exchange between contacts with MavC, Ube2N, and/or solvent. Please see below (response to the comment after the next one). We have made edits to the text to clearly discuss that the NMR and BLI data together suggest that binding is driven by Ube2N-MavC interactions as follows:

“BLI and NMR experiments show MavC readily binds Ube2N-SS-Ub. BLI binding titrations yield a K_d of $\sim 2.4 \mu\text{M}$ for Ube2N-SS-Ub, nearly the same affinity as observed for free Ube2N (Fig. 2a), consistent with binding dominated by interactions with Ube2N.”

- In previous studies, free Ub and even a Ub G75A/G76A mutant that cannot form a thioester intermediate with Ube2N were shown to be linked to Ube2N by MavC. The authors should discuss these results.

In the Gan et al. Nature Microbiology paper, the reactivity of WT Ub and Ub G75A/G76A were compared by immunoblot. While it is true that the mutant, which cannot form a thioester intermediate, did undergo a small degree of crosslinking, the WT Ub showed significantly more formation of Ub-Ube2N product. This result appears to be in line with our study, and we have included this statement in our manuscript as follows:

“These results are also in line with observations of Gan et al, which show that a mutant of Ub lacking the last two glycines is modified to a significantly lower extent than wild type Ub.”

- The quality of the Ube2N-thioester linked Ub 15N spectra in Fig. 2d is quite poor and contains many signals with low intensities. Are these due to chemical exchange or due to impurities or degradation of the sample? It would be helpful if the authors showed an overlay of the 15N

spectra of free Ub with the Ube2N-SS-linked Ub in the Supplement and labeled the Ub resonances. Do the CSPs in Fig. 2d agree with the Ub binding surfaces of the crystal structures?

We appreciate the feedback and can point to two explanations for the intensity loss: 1) Overall intensity loss due to the large molecular weight of the complex. 2) Many, but not all, resonances are lost due to exchange broadening.

We have accordingly updated the text as follows:

“In the conjugate, not all Ub resonances are observed or of equal intensity. This is due to the Ub subunit alternating between open states, where Ub makes few contacts with the E2, and closed states, where the Ub subunit is in close contact with the E2.²⁸ This equilibrium results in exchange broadening of resonances whose environments differ in the open and closed states. Ub resonances that remain and largely overlap with those of free Ub can be assigned by inspection (**Fig. SX**).

In marked contrast to the addition of MavC to free Ub (**Fig. 2c**), large perturbations in the spectrum of the Ub subunit of ²H, ¹⁵N-Ub-SS-Ube2N are now observed upon formation of a ²H, ¹⁵N-Ub-SS-Ube2N/MavC complex (**Fig. 2d**). An overall loss in peak intensity is observed consistent with the large increase in molecular weight (~65 kDa) upon complex formation. In addition, a number of Ub resonances disappear. Again, this behavior can be attributed to resonance exchange broadening where a subset of Ub residues exchange between contacts with MavC, Ube2N, and/or solvent. Thus, in solution, the Ub subunit is not rigidly fixed to the enzyme active. However, the high local concentration of Ub provided by MavC binding of the Ube2N~Ub conjugate significantly increases observed contacts.”

- Fig. 2e: The high concentration of DTT (1 mM) in the reaction buffer compared to the Ube2N~Ub disulfide (25 μM) should reduce the disulfide to free Ube2N and Ub. The authors should show loading controls in the absence of MavC for each time point to demonstrate the integrity of the Ube2N~Ub disulfide during the reaction.

Thank you for pointing this out; we mistakenly wrote that DTT was included in the buffer for that experiment, when in fact it was not. We have rectified this error in the Methods section and have also included a loading control of the Ube2N-SS-Ub as requested. We find that in a non-reducing gel, the disulfide conjugate remained stable throughout the course of the experiment.

We have updated Figure 2 accordingly.

- Fig. 5e,f: Although the YYE mutant is virtually incapable of performing transglutamination, it still effectively down-regulates NF-κB compared to the C74A mutant. The authors should discuss this result. Fig. 5f: The MavC gel (six lanes) does not match with the Tubulin gel (seven lanes). What does the bar below the bar chart refer to? Significance should be tested in the bar

chart and the gels should be labeled. The methods for measuring NF- κ B activity should be described in more detail. The error bars and number of experiments are not described.

Thank you for pointing this out. We agree that the triple mutant is not completely inhibiting NF- κ B signaling under the conditions of our experiment described in the original Fig. 5E/5F. To address this, we designed more aggressive, quadruple mutants (YYE+M317D, predicted to further disrupt Ube2N binding and YYE + W255A, predicted to further disrupt Ub binding). These mutants were studied in both enzymatic activity assays using higher enzyme concentrations and were found to be less active than the original YYE triple mutant (Found in Figure S3c). When examined under conditions of infection, these mutants show a drastic defect in ubiquitination of Ube2N (Figure 5E), almost to the extent observed with the catalytically inactive C74A mutant. These mutants were also examined for inhibition of NF- κ B signaling after TRAF6 overexpression (Figure 5F). These show that inhibition of substrate binding lead to a defect in the immunomodulatory effects of MavC.

We have revised Fig. 5 to include these new experiments. In these new figures, the loading control lanes correspond with the experimental lanes, and we include error bars and number of experiments.

The methods for measuring NF- κ B signaling have been described in further detail in the Methods section.

- The paper falls short of analyzing the conformation of the Ube2N~Ub thioester and Ube2N-Ub complex. The authors should compare / superimpose their structures with known structures of Ube2N (or Ubc13)~Ub conjugates. Would E3 binding clash with the MavC interaction? A more thorough analysis of the Ube2N-Ub isopeptide would be helpful to understand the mechanism underlying E2 inhibition and NF- κ B inactivation. Does the structure of the product complex provide any cues in this direction? A CSP analysis of ¹⁵N Ub isopeptide linked to Ube2N might be useful in this respect.

Thank you for the feedback. We have compared the available Ube2N~Ub containing structures in the PDB and aligned the Ube2N/Ubc13 proteins with the reported MavC Ube2N-Ub product. We include this analysis in Supplementary Fig. S5D. Indeed, the conformation adopted by Ub-Ube2N when bound to MavC resembles a subset of the Ube2N~Ub structures as indicated in Fig. S5D. We speculate that the covalent attachment of Ub to Lys92 of Ube2N causes a steric occlusion, preventing re-charging of Ube2N by E1. The product-bound complex supports this model of inhibition. Indeed, we have found that the MavC product, Ube2N-Ub, fails to become charged by E1. We have included this result in Supplementary Figure S5E.

Regarding E3 binding, we do predict that E3 binding to Ube2N would clash with the MavC interaction. Further, the affinity of MavC to Ube2N is much greater than that of most RING E3

ligases (~10-100 fold). We have clarified our statement in the text to better explain this and cite Nishide et al. J. Mol. Biol. 2013. (ref. 25) as follows:

“This is the same Ube2N surface shown to interact with numerous eukaryotic E3 ligases²⁵”

Additionally, we have the following text in the Discussion section:

“Though other E2s harbor a structurally equivalent target lysine residue, selectivity for Ube2N is achieved by binding the same interface recognized by cognate eukaryotic E3 ligases (**Fig. S3b-d**).”

“This inhibition may occur due to blocking of re-charging of Ube2N by the E1 enzyme, and also by competitive displacement of E3 binding.”

We agree that the CSP analysis of Ub-Ube2N would be useful in probing the dynamics of the MavC product. However, in light of the experimental data given above showing that the crosslinked product cannot be charged by E1, we do not believe that the CSP analysis, a considerable undertaking at this stage, would add any substantial new information to alter the main conclusions pertaining to this point.

- The subsection describing the role of Uev1a is somewhat preliminary. The authors should therefore move Fig. 7a-b to the Supplement and substantiate their speculations by adding a structural model of the Ube2N-Ub:MavC complex including Uev1a. The units in Fig. 7a for the time points are missing.

We have accordingly moved Fig. 7a to Figure S5B and have also generated a model of Ube2N-Ub:MavC with Uev1A, included as Figure S5C. The model suggests that there is no conflict between Uev1A and MavC binding to Ube2N and we have updated the text as follows:

“From our structural analysis, Uev1A binding to Ube2N~Ub is unlikely to interfere with MavC binding. (**Fig. S5c**).”

The units in Figure S5B are concentration values rather than time values, and we have updated the figure to reflect that more clearly.

- Fig. S5 nicely summarizes the structural data. I would therefore recommend replacing Fig. 7a,b with Fig. S5. The Ube2N C87 and K92, Ub Q40 and MavC C74 side chains should be labeled. The isopeptide linked Ube2N-Ub / product complex should be shown as well. It would also be helpful to include these cartoons on top of the structures shown in Fig. 6a-c.

We appreciate the feedback, and accordingly have moved Figure S5 to Figure 7A with the suggested changes: Residues have been labeled and the product bound complex is also shown. We feel that Fig. 7B is a necessary summary of the biological relevance of MavC and therefore we wish to retain it.

We appreciate the suggestion, while the cartoons shown in Fig. 7A capture the essence of MavC's dynamic movements, it is difficult from our structural data to relate each overall motion

of the INS domain with unfolding of the 3-10 helix, which is the main focus of Fig. 6a-c. We therefore feel, for clarity's sake it would be better not to place the cartoons with the panels in Fig. 6.

Minor issues:

- The formatting of Greek letters and superscripts is missing / has been lost at several positions. A single font type should be used for all figures.

We have made changes accordingly to the figures and text to ensure consistency.

- Throughout the text, the authors should make clear which catalytic Cys / thioester is referred to, Ube2N or MavC.

We have clarified in the text the nature of the catalytic Cys where necessary.

- Ube2N is also referred to as Ube2N and UBE2N. The spelling should be consistent throughout the paper. Similarly, a space should always be included between values and their units.

We have ensured the spelling is consistent throughout.

- Fig. S2: Fig. S2d: The ^{15}N axis should be labeled, since it should be different from Fig. S2c (HSQC vs. TROSY spectrum). Fig. S2e and f: The corresponding NMR spectra should be shown. What is the CSP cut-off used in Fig. 2f? Fig. S2h: The units for the time points are missing.

Thanks for the feedback. Separate axes are shown for the HSQC and TROSY spectra in Fig. S2. Corresponding NMR spectra for the graphs have also been inserted. The CSP cut-off used in S2f is one standard deviation above the mean. We have also included the time points for the deamidation experiment.

- Fig. S3: The numbering of the panels is not consistent with the text, e.g. Fig. S3e is mentioned in the text before Fig. S3a. Fig. S3a: The YYE mutant should be spelled out in the same way as the other mutants. Fig. S3c: The units for the time points are missing. Fig. S3e: The gel is not described in the legend.

Thank you for pointing that out, we have accordingly moved Fig. S3e to Fig. S3a to resolve the inconsistency. Due to limitations on space in the figure, instead of spelling out the YYE mutant we decided to provide its full name in the figure caption.

- Fig. 4c: The figure legend should be more explicit. How are the Ub species visualized? What

does the “Ctrl” lane refer to?

Thanks for the feedback, we have made these requested changes in Fig. 4c. The “Ctrl” lane refers to Ub without any MavC added. We clarified that in the caption.

- Fig. S4d: The figure legend should be more explicit. “corresponds to maximum rmsd” of what?

We have accordingly changed the caption for Fig. S4d, stating “maximum rmsd relative to the calculated average structure of Ube2N.”

- Constructs and purification of E2 enzymes other than Ube2N and of Cif are not described.

We have included purification of the other E2s and Cif in the Methods section. Briefly, the E2s were purified using GST affinity chromatography, similarly to MavC, and Cif was purified using nickel affinity chromatography, similarly to Ube2N.

- Software used for multiple sequence alignments is not described.

We accordingly have included the software name (Clustal Omega) along with the relevant reference.

Reviewers' Comments:

Reviewer #1:

Remarks to the Author:

The authors have addressed my concerns. Although they hope to use the crystallographic statistics they reported in the old manuscript, I think it is OK, given that the final structural model is the same. The manuscript is now suitable for publication.